# MONOTONIC KRONECKER-FACTORED LATTICE

**Nobuyuki Morioka**,[*] **Erez Louidor**,[*] **William Bakst**[*]
Google Research
{nmorioka,erez,wbakst}@google.com

## ABSTRACT

It is computationally challenging to learn flexible monotonic functions that guarantee model behavior and provide interpretability beyond a few input features, and in a time where minimizing resource use is increasingly important, we must be able to learn such models that are still efficient. In this paper we show how to effectively and efficiently learn such functions using Kronecker-Factored Lattice (KFL), an efficient reparameterization of flexible monotonic lattice regression via Kronecker product. Both computational and storage costs scale linearly in the number of input features, which is a significant improvement over existing methods that grow exponentially. We also show that we can still properly enforce monotonicity and other shape constraints. The KFL function class consists of products of piecewise-linear functions, and the size of the function class can be further increased through ensembling. We prove that the function class of an ensemble of $M$ base KFL models strictly increases as $M$ increases up to a certain threshold. Beyond this threshold, every multilinear interpolated lattice function can be expressed. Our experimental results demonstrate that KFL trains faster with fewer parameters while still achieving accuracy and evaluation speeds comparable to or better than the baseline methods and preserving monotonicity guarantees on the learned model.

## 1 INTRODUCTION

Many machine learning problems have other requirements in addition to accuracy, such as efficiency, storage, and interpretability. For example, the ability to learn flexible monotonic functions at scale is useful in practice because machine learning practitioners often know apriori which input features positively or negatively relate to the output and can incorporate these hints as inductive bias in the training to further regularize the model (Abu-Mostafa, 1993) and guarantee its expected behavior on unseen examples. It is, however, computationally challenging to learn such functions efficiently. In this paper, we extend the work of interpretable monotonic lattice regression (Gupta et al., 2016) to significantly reduce computational and storage costs without compromising accuracy.

While a linear model with nonnegative coefficients is able to learn simple monotonic functions, its function class is restricted. Prior works proposed non-linear methods (Sill, 1997; Dugas et al., 2009; Daniels & Velikova, 2010; Qu & Hu, 2011) to learn more flexible monotonic functions, but they are shown to work only in limited settings of small datasets and low dimensional feature spaces. Monotonic lattice regression (Gupta et al., 2016), an extension of lattice regression (Garcia et al., 2012), learns an interpolated look-up table with linear inequality constraints that impose monotonicity. This has been demonstrated to work with millions of training examples and achieve competitive accuracy against, for example, random forests; however, because the number of model parameters scales exponentially in the number of input features, it is difficult to apply such models in high dimensional feature space settings. To overcome this limitation, (Canini et al., 2016) incorporated ensemble learning to combine many tiny lattice models, each capturing non-linear interactions among a small random subset of features.

This paper proposes Kronecker-Factored Lattice (KFL), a novel reparameterization of monotonic lattice regression via Kronecker product to achieve significant parameter efficiency and simultaneously provide guaranteed monotonicity of the learned model with respect to a user-prescribed set of input

---

[*]Equal contribution.

features for user trust. Both inference and storage costs scale linearly in the number of features. Hence, it potentially allows for more features to non-linearly interact. Kronecker factorization has been applied to a wide variety of problems including optimization (Martens & Grosse, 2015; George et al., 2018; Osawa et al., 2019), convolutional neural networks (Zhang et al., 2015), and recurrent neural networks (Jose et al., 2018), but, to the best of our knowledge, has not yet been explored to learn flexible monotonic functions.

The main contributions of this paper are: (1) reparameterization of monotonic lattice regression to achieve linear evaluation time and storage of the resulting subclass of models, (2) proving sufficient and necessary conditions for a KFL model to be monotonic as well as convex and nonnegative, (3) showing how the conditions for monotonicity can be efficiently imposed during training, (4) characterizing the values of $M$ for which the capacity of the function class of an ensemble of $M$ base KFL models strictly increases and showing that with a large enough $M$ an ensemble can express every multilinear interpolated lattice function, and (5) providing experimental results on both public and proprietary datasets that demonstrate that KFL has accuracy and evaluation speeds comparable to or better than other lattice methods while requiring less training time and fewer parameters.

## 2 NOTATION AND OVERVIEW OF MONOTONIC LATTICE REGRESSION

For $n \in \mathbb{N}$ we denote by $[n]$ the set $\{1, 2, \ldots, n\}$. For $\mathbf{x} \in \mathbb{R}^D$ and $d \in [D]$, let $\mathbf{x}[d]$ denote the $d$th entry of $\mathbf{x}$. We use $\mathbf{e}_{d,n}$ to denote the one-hot vector in $\{0,1\}^n$ where $\mathbf{e}_{d,n}[j] = 1$ if $j = d$ and $\mathbf{e}_{d,n}[j] = 0$, otherwise. When $n$ is clear from the context we write $\mathbf{e}_d$. For two real vectors $\mathbf{v}, \mathbf{w}$ of the same length we use $\mathbf{v} \cdot \mathbf{w}$ to denote their dot product. We write $\mathbf{v} \leq \mathbf{w}$ and $\mathbf{v} \geq \mathbf{w}$ if the respective inequality holds entrywise. We denote by $\mathbf{0}$ the zero vector in a real vector space whose dimension is clear from the context. For two sets $S$ and $T$ we use $S \times T$ to denote their Cartesian product and $S \setminus T$ to denote the set of elements in $S$ but not in $T$. We denote by $|S|$ the cardinality of $S$. For $\mathbf{w} \in \mathbb{R}^V$, we use $f_{\mathrm{pwl}}(x; \mathbf{w})$ to denote the 1-dimensional continuous piecewise linear function $f_{\mathrm{pwl}}(\cdot; \mathbf{w}) : [0, V-1] \to \mathbb{R}$ whose graph has $V-1$ linear segments, where for $i = 1, \ldots, V-1$, the $i$th segment connects the points $(i-1, \mathbf{w}[i])$ with $(i, \mathbf{w}[i+1])$. See Figure 1 for an example. Observe that for any $\alpha, \beta \in \mathbb{R}$ and vectors $\mathbf{w}, \mathbf{v} \in \mathbb{R}^V$, we have $f_{\mathrm{pwl}}(\cdot; \alpha\mathbf{w} + \beta\mathbf{v}) = \alpha f_{\mathrm{pwl}}(\cdot; \mathbf{w}) + \beta f_{\mathrm{pwl}}(\cdot; \mathbf{v})$.

| Symbol | Description |
|---|---|
| $D$ | Number of features |
| $\mathbf{e}_{d,n}$ | "1-hot" $n$-dimensional vector with 1 in the $d$th entry |
| $f_{\mathrm{pwl}}$ | 1-dimensional piecewise linear function |
| $f_{\mathrm{mll}}$ | Multilinear lattice function |
| $f_{\mathrm{sl}}$ | Simplex lattice function |
| $f_{\mathrm{kfl}}$ | KFL function |
| $\mathcal{V}$ | $D$-dimensional size of lattice |
| $\mathcal{M}_\mathcal{V}$ | Set of vertices of lattice of size $\mathcal{V}$ |
| $\mathcal{D}_\mathcal{V}$ | Domain of a lattice model |
| $\boldsymbol{\Phi}$ | Interpolation kernel. |
| $\boldsymbol{\theta}$ | Vector of lattice parameters |
| $\otimes_i \mathbf{v}_i$ | Outer / Kronecker product of vectors $\mathbf{v}_i$ |
| KFL($\mathcal{V}$,$M$) | Class of ensembles of $M$ KFL functions of size $\mathcal{V}$ |
| $r(T)$ | Rank of tensor $T$ |

Table 1: Table of major notation

Lattice models, proposed in Garcia et al. (2012), are interpolated look-up tables such that the function parameters are the function values sampled on a regular grid. They have been shown to exhibit efficient training procedures such that the resulting model is guaranteed to satisfy various types of shape constraints, including monotonicity and convexity (Gupta et al., 2016; 2018; Cotter et al., 2019), without severely restricting the model class. See Figure 2 for an example.

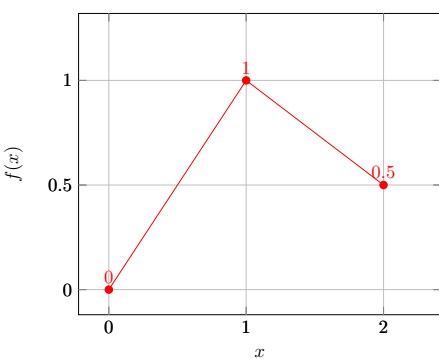
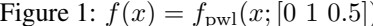

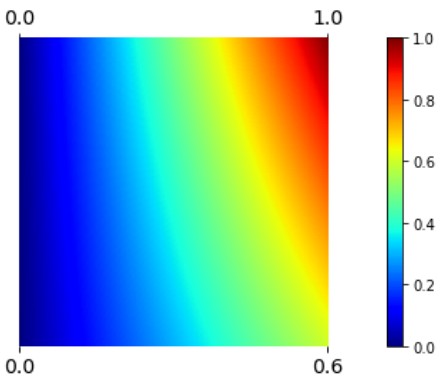

Figure 1: $f(x) = f_{\text{pwl}}(x; [0\ 1\ 0.5])$

Figure 2: $2\times2$ multilinear interpolated lattice with four parameters defining a monotonically increasing function in both dimensions.

More precisely, a $D$-dimensional lattice of size $\mathcal{V}\in\mathbb{N}^D$ consists of a regular $D$-dimensional grid of look-up table vertices $\mathcal{M}_\mathcal{V} = \{0, 1, \ldots, \mathcal{V}[1]-1\} \times \ldots \times \{0, 1, \ldots, \mathcal{V}[D]-1\}$. Thus, $\mathcal{V}[d]$ is the number of vertices in the lattice in the $d$th dimension, and the grid has $\prod_{d=1}^D \mathcal{V}[d]$ vertices. We denote by $\mathcal{D}_\mathcal{V} \in \mathbb{R}^D$ the domain of the lattice model given by $\mathcal{D}_\mathcal{V} = [0, V[1] - 1] \times \ldots \times [0, V[D] - 1]$. Typically, each feature $d$ is "calibrated" so that its value lies in $[0, \mathcal{V}[d]-1]$, using the piecewise-linear calibration technique of (Gupta et al., 2016) (see also Section 3.5). The lattice parameters are a vector $\boldsymbol{\theta} \in \mathbb{R}^{\mathcal{M}_\mathcal{V}}$, where $\mathbb{R}^{\mathcal{M}_\mathcal{V}}$ denotes the set of vectors of length $|\mathcal{M}_\mathcal{V}|$ with entries indexed by $\mathcal{M}_\mathcal{V}$: for each vertex $\mathbf{v} \in \mathcal{M}_\mathcal{V}$, there is a corresponding parameter we denote by $\theta_\mathbf{v}$. The *lattice model* or *function* is obtained by *interpolating* the parameters using one of many possible interpolation schemes specified by an interpolation kernel $\boldsymbol{\Phi} : \mathcal{D}_\mathcal{V} \to \mathbb{R}^{\mathcal{M}_\mathcal{V}}$. Given $\boldsymbol{\Phi}$, the resulting lattice model $f : \mathcal{D}_\mathbf{v} \to \mathbb{R}$ is defined by

$$f(\mathbf{x}; \boldsymbol{\theta}) = \boldsymbol{\theta} \cdot \boldsymbol{\Phi}(\mathbf{x}) = \sum_{\mathbf{v}\in\mathcal{M}_\mathcal{V}} \theta_\mathbf{v} \big(\boldsymbol{\Phi}(\mathbf{x})\big)_\mathbf{v}, \tag{1}$$

where $(\boldsymbol{\Phi}(\mathbf{x}))_\mathbf{v}$ denotes the entry of $\boldsymbol{\Phi}(\mathbf{x})$ with index $\mathbf{v}$. Typically, one requires that for all $\mathbf{u}, \mathbf{v} \in \mathcal{M}_\mathcal{V}$, $(\boldsymbol{\Phi}(\mathbf{v}))_\mathbf{u} = 1$ if $\mathbf{u} = \mathbf{v}$, and 0 otherwise, so that $f(\mathbf{v}) = \theta_\mathbf{v}$. In Gupta et al. (2016) the authors present two interpolation schemes termed 'multilinear' and 'simplex'. We denote the corresponding kernels by $\boldsymbol{\Phi}_\mathcal{V}^{\text{multilinear}}$ and $\boldsymbol{\Phi}_\mathcal{V}^{\text{simplex}}$ and the resulting lattice models by $f_{\text{mll}}(\cdot; \boldsymbol{\theta})$ and $f_{\text{sl}}(\cdot; \boldsymbol{\theta})$, respectively. We give the definition of $\boldsymbol{\Phi}_\mathcal{V}^{\text{multilinear}}$ here since we make use of it later, but refer the reader to (Gupta et al., 2016) for the definition of simplex interpolation.

$$\big(\boldsymbol{\Phi}_\mathcal{V}^{\text{multilinear}}(\mathbf{x})\big)_\mathbf{v} = \prod_{j\in[D]} f_{\text{pwl}}(\mathbf{x}[j]; \mathbf{e}_{\mathbf{v}[j]+1, \mathcal{V}[j]}), \ \mathbf{v} \in \mathcal{M}_\mathcal{V} \tag{2}$$

For both multilinear and simplex interpolated lattice functions, the resulting function is increasing (resp. decreasing) in a given direction if and only if the function is increasing (resp. decreasing) on the grid's vertices in that direction (Gupta et al., 2016, Lemmas 1, 3). As a result, training a lattice model with these interpolation schemes to respect the monotonicity constraint can be done by solving a constrained optimization problem with linear inequality constraints (Gupta et al., 2016).

Evaluating the lattice function at a single point can be done in $\mathcal{O}(2^D)$ time for multilinear interpolation and in $\mathcal{O}(D \log D)$ time for simplex interpolation (Gupta et al., 2016). Both interpolation schemes require $\mathcal{O}(\prod_d \mathcal{V}[d])$ space for storing the parameters (assuming a constant number of bits per parameter). In the next section we present a sub-class of the set of multilinear-interpolation lattice functions, which we term KFL($\mathcal{V}$). Each function in this class can be evaluated in $\mathcal{O}(D)$ time and requires only $\mathcal{O}(\sum_d \mathcal{V}[d])$ space for its parameters. Moreover, like the multilinear and simplex interpolated lattice function classes, there is a necessary and sufficient condition for a function in KFL($\mathcal{V}$) to be monotonic in a subset of its inputs, and this condition can be efficiently checked. As a result, one can train models in KFL($\mathcal{V}$) that are guaranteed to be monotonic in a prescribed set of features.

## 3 KRONECKER-FACTORED LATTICE

The *Kronecker product* of two matrices $\mathbf{A} = (a_{i,j}) \in \mathbb{R}^{m \times n}$ and $\mathbf{B} = (b_{i,j}) \in \mathbb{R}^{p \times q}$ is defined as the $mp \times nq$ real block matrix given by:

$$\mathbf{A} \otimes \mathbf{B} = \begin{bmatrix} a_{11}\mathbf{B} & \cdots & a_{1n}\mathbf{B} \\ \vdots & \ddots & \vdots \\ a_{m1}\mathbf{B} & \cdots & a_{mn}\mathbf{B} \end{bmatrix}. \tag{3}$$

The Kronecker product is associative and generally non-commutative; however, $\mathbf{A} \otimes \mathbf{B}$ and $\mathbf{B} \otimes \mathbf{A}$ are permutation equivalent. We extend the definition of Kronecker product to vectors by regarding them as matrices with a single row or column. For $s$ vectors $\mathbf{v}_i \in \mathbb{R}^{n_i}$, we use the notation $\otimes_{i=1}^{s} \mathbf{v}_i = \mathbf{v}_1 \otimes \ldots \otimes \mathbf{v}_s$. Let $\mathcal{I} = \{0, \ldots, n_1 - 1\} \times \ldots \times \{0, \ldots, n_s - 1\}$. We index the entries of $\otimes_{i=1}^{s} \mathbf{v}_i$ by $\mathcal{I}$, so that the entry with index $(i_1, \ldots, i_s) \in \mathcal{I}$ is $\prod_{j=1}^{s} \mathbf{v}_j[i_j + 1]$.

The Kronecker product satisfies the *mixed-product property*: for 4 matrices $\mathbf{A} \in \mathbb{R}^{m \times n}, \mathbf{B} \in \mathbb{R}^{p \times q}$, $\mathbf{C} \in \mathbb{R}^{n \times r}, \mathbf{D} \in \mathbb{R}^{q \times s}$ it holds that $(\mathbf{A} \otimes \mathbf{B})(\mathbf{C} \otimes \mathbf{D}) = (\mathbf{AC}) \otimes (\mathbf{BD})$, where for two matrices $M, N$, we denote their conventional product by $MN$. Consequently for vectors $\mathbf{a}, \mathbf{c} \in \mathbb{R}^n$ and $\mathbf{b}, \mathbf{d} \in \mathbb{R}^q$ it holds that

$$(\mathbf{a} \otimes \mathbf{b}) \cdot (\mathbf{c} \otimes \mathbf{d}) = (\mathbf{a} \cdot \mathbf{c}) \otimes (\mathbf{b} \cdot \mathbf{d}) = (\mathbf{a} \cdot \mathbf{c})(\mathbf{b} \cdot \mathbf{d}), \tag{4}$$

where the rightmost equality follows since Kronecker product of scalars reduces to a simple product. See (Loan, 2000) for more details on Kronecker products.

We are now ready to define the class $\text{KFL}(\mathcal{V})$. Let $\mathcal{V} \in \mathbb{N}^D$ and fix a lattice of size $\mathcal{V}$. The class $\text{KFL}(\mathcal{V})$ is the subclass of all multilinear interpolation lattice functions whose parameters are Kronecker products of $D$ factors. Precisely,

$$\text{KFL}(\mathcal{V}) = \{f_{\text{mll}}(\cdot; \otimes_{d=1}^{D} \mathbf{w}_d) : \forall d \; \mathbf{w}_d \in \mathbb{R}^{\mathcal{V}[d]}\}.$$

We reparameterize the functions in KFL and use $f_{\text{kfl}}(\cdot; \mathbf{w}_1, \ldots, \mathbf{w}_d) = f_{\text{mll}}(\cdot; \otimes_{d=1}^{D} \mathbf{w}_d)$ to denote the function with parameters $\{\mathbf{w}_d\}_d$. The following proposition shows that functions in $\text{KFL}(\mathcal{V})$ are products of 1-dimensional piecewise linear functions. It essentially follows from the fact that $\boldsymbol{\Phi}_{\mathcal{V}}^{\text{multilinear}}(\mathbf{x})$ can be expressed as a Kronecker product of $D$ vectors.

**Proposition 1.** *Let $\mathcal{V} \in \mathbb{N}^D$. For all $\mathbf{w}_1 \in \mathbb{R}^{\mathcal{V}[1]}, \ldots, \mathbf{w}_D \in \mathbb{R}^{\mathcal{V}[D]}$*

$$f_{\text{kfl}}(\mathbf{x}; \mathbf{w}_1, \ldots, \mathbf{w}_D) = \prod_d f_{\text{pwl}}(\mathbf{x}[d]; \mathbf{w}_d), \quad \text{for all } \mathbf{x} \in \mathcal{D}_{\mathcal{V}}. \tag{5}$$

*Proof of Proposition 1.* For $v \in \mathbb{N}$, let $\boldsymbol{\Psi}_v : \mathbb{R} \to \mathbb{R}^v$ be given by $\boldsymbol{\Psi}_v(x) = [f_{\text{pwl}}(x; \mathbf{e}_{1,v}) \; f_{\text{pwl}}(x; \mathbf{e}_{2,v}) \; \ldots \; f_{\text{pwl}}(x; \mathbf{e}_{v,v})]$. Let $\mathbf{w}_1, \ldots, \mathbf{w}_D$ be vectors satisfying the conditions in the proposition. By the definition of $\text{KFL}(\mathcal{V})$, we have $f_{\text{kfl}}(\cdot; \mathbf{w}_1, \ldots, \mathbf{w}_D) = f_{\text{mll}}(\cdot; \otimes_{d=1}^{D} \mathbf{w}_d)$. Fix $\boldsymbol{\Phi} = \boldsymbol{\Phi}_{\mathcal{V}}^{\text{multilinear}}$. It follows from (2) that for all $\mathbf{x} \in \mathcal{D}_{\mathcal{V}}$ we have $\boldsymbol{\Phi}(\mathbf{x}) = \otimes_{d=1}^{D} \boldsymbol{\Psi}_{\mathcal{V}[d]}(\mathbf{x}[d])$. Therefore using (4) we get, for all $\mathbf{x} \in \mathcal{D}_v$,

$$f_{\text{kfl}}(\mathbf{x}; \mathbf{w}_1, \ldots, \mathbf{w}_D) = f_{\text{mll}}(\mathbf{x}; \otimes_{d=1}^{D} \mathbf{w}_d) = (\otimes_{d=1}^{D} \mathbf{w}_d) \cdot (\otimes_{d=1}^{D} \boldsymbol{\Psi}_{\mathcal{V}[d]}(\mathbf{x}[d]))$$
$$= \prod_{d=1}^{D} (\mathbf{w}_d \cdot \boldsymbol{\Psi}_{\mathcal{V}[d]}(\mathbf{x}[d])). \tag{6}$$

Now, for each $d \in [D]$ we have

$$\mathbf{w}_d \cdot \boldsymbol{\Psi}_{\mathcal{V}[d]}(\mathbf{x}[d]) = \sum_{i \in [\mathcal{V}[d]]} \mathbf{w}_d[i] f_{\text{pwl}}(\mathbf{x}[d]; \mathbf{e}_{i,\mathcal{V}[d]}) = f_{\text{pwl}}\left(\mathbf{x}[d]; \sum_i \mathbf{w}_d[i]\mathbf{e}_{i,\mathcal{V}[d]}\right)$$
$$= f_{\text{pwl}}(\mathbf{x}[d]; \mathbf{w}_d). \tag{7}$$

Substituting (7) into (6), we obtain (5). $\qquad\square$

It follows from Proposition 1 that evaluating a function in $\text{KFL}(\mathcal{V})$ requires evaluating $D$ piecewise linear functions with uniformly spaced knots and computing the product. This can be done in time $\mathcal{O}(D)$. The storage complexity is $\mathcal{O}(\# \text{ parameters}) = \mathcal{O}(\sum_i \mathcal{V}[i])$.

## 3.1 MONOTONICITY CRITERIA FOR KFL

Let $i \in \{1, \ldots, D\}$. We say that a function $f : \mathcal{D} \to \mathbb{R}$, where $\mathcal{D} \subseteq \mathbb{R}^D$, is increasing (resp. decreasing) in direction $i$ in $\mathcal{D}$, if for every $\mathbf{x} \in \mathcal{D}$ and real $\delta \geq 0$, such that $\mathbf{x} + \delta \mathbf{e}_i \in \mathcal{D}$, it holds that $f(\mathbf{x}) \leq f(\mathbf{x} + \delta \mathbf{e}_i)$ (resp. $f(\mathbf{x}) \geq f(\mathbf{x} + \delta \mathbf{e}_i)$). Note that a function that does not depend on $\mathbf{x}[i]$ is thus "trivially" increasing in direction $i$. In this section we disregard such edge cases and require a function increasing in the $i$th direction to strictly increase on at least one pair of inputs. To simplify the exposition, we only deal with increasing functions in this section–all the results transfer naturally to the decreasing case, as well.

The next proposition shows a sufficient and necessary condition for a function in $\mathrm{KFL}(\mathcal{V})$ to be increasing in a subset of its features in $\mathcal{D}_{\mathcal{V}}$. In Section 3.5 we explain how to use this result to train models that are guaranteed to be monotonic.

**Proposition 2.** *Fix $\mathcal{V} \in \mathbb{N}^D$, let $f \in \mathrm{KFL}(\mathcal{V})$ and let $i_1, \ldots, i_p \in [D]$ be distinct direction indices for $p \leq D$. Then $f$ is increasing in directions $i_1, \ldots, i_p$ iff $f$ can be written as:*

$$f(\mathbf{x}) = \sigma \prod_d f_{\mathrm{pwl}}(\mathbf{x}[d]; \mathbf{w}_d), \tag{8}$$

*where $\sigma \in \{+1, -1\}$, and $\mathbf{w}_d \in \mathbb{R}^{\mathcal{V}[d]}, d \in [D]$ are vectors satisfying the following 3 conditions:*

  1. *If $p \geq 2$ then $\mathbf{w}_d \geq \mathbf{0}$ for all $d \in [D]$.*

  2. *If $p = 1$ then $\mathbf{w}_d \geq \mathbf{0}$ for all $d \in [D] \setminus \{i_1\}$.*

  3. *For all $j \in [p]$, $\sigma \mathbf{w}_{i_j}[1] \leq \sigma \mathbf{w}_{i_j}[2] \leq \ldots \leq \sigma \mathbf{w}_{i_j}[\mathcal{V}[i_j]]$.*

See Appendix A for the proof.

## 3.2 CRITERIA FOR OTHER SHAPE CONSTRAINTS

While the focus of this paper is on learning flexible monotonic functions, other shape constraints, namely convexity and nonnegativity, can easily be imposed on KFL. This section presents the propositions showing sufficient and necessary conditions for these constraints. More shape constraints beyond these are left for future work.

We say that a function $f : \mathcal{D} \to \mathbb{R}$ is convex (resp. concave) in direction $i$ in $\mathcal{D}$ if for every $\mathbf{x} \in \mathcal{D}$ and real $\delta \geq 0$ such that $\mathbf{x}, \mathbf{x} + \delta \mathbf{e}_i \in \mathcal{D}$, we have $f(t\mathbf{x} + (1-t)(\mathbf{x} + \delta \mathbf{e}_i)) \leq tf(\mathbf{x}) + (1-t)f(\mathbf{x} + \delta \mathbf{e}_i)$ (resp. $f(t\mathbf{x} + (1-t)(\mathbf{x} + \delta \mathbf{e}_i)) \geq tf(\mathbf{x}) + (1-t)f(\mathbf{x} + \delta \mathbf{e}_i)$) for all $t \in [0, 1]$. As with monotonicity, we require a function convex (resp. concave) in direction $i$ to strictly satisfy the inequality for some $\mathbf{x}, \delta$ and $t$ (i.e. we disregard functions that are affine in direction $i$ in $\mathcal{D}$). The following proposition shows sufficient and necessary criteria for a function to be convex in a subset of directions; an analogous criteria can be derived for concavity.

**Proposition 3.** *Fix $\mathcal{V} \in \mathbb{N}^D$, let $f \in \mathrm{KFL}(\mathcal{V})$ and let $i_1, \ldots, i_p \in [D]$ be distinct direction indices for $p \leq D$. Then $f$ is convex in directions $i_1, \ldots, i_p$ iff $f$ can be written as:*

$$f(\mathbf{x}) = \sigma \prod_d f_{\mathrm{pwl}}(\mathbf{x}[d]; \mathbf{w}_d), \tag{9}$$

*where $\sigma \in \{+1, -1\}$, and $\mathbf{w}_d \in \mathbb{R}^{\mathcal{V}[d]}, d \in [D]$ are vectors satisfying the following 3 conditions:*

  1. *If $p \geq 2$ then $\mathbf{w}_d \geq \mathbf{0}$ for all $d \in [D]$.*

  2. *If $p = 1$ then $\mathbf{w}_d \geq \mathbf{0}$ for all $d \in [D] \setminus \{i_1\}$.*

  3. *For all $j \in [p]$, $\mathcal{V}[i_j] = 2$ or $\mathcal{V}[i_j] > 2$ and $\sigma(\mathbf{w}_{i_j}[k+2] - \mathbf{w}_{i_j}[k+1]) \geq \sigma(\mathbf{w}_{i_j}[k+1] - \mathbf{w}_{i_j}[k])$ for all $k \in [\mathcal{V}[i_j] - 2]$.*

See Appendix A for the proof.

**Proposition 4.** *Fix $\mathcal{V} \in \mathbb{N}^D$, let $f \in \mathrm{KFL}(\mathcal{V})$. Then $f$ is nonnegative (resp. strictly positive) iff $f$ can be written as $f(\mathbf{x}) = \prod_d f_{\mathrm{pwl}}(\mathbf{x}[d]; \mathbf{w}_d)$, where $\mathbf{w}_d \geq \mathbf{0}$ (resp. $\mathbf{w}_d > \mathbf{0}$) for all $d \in [D]$.*

See Appendix A for the proof.

## 3.3 ENSEMBLES OF KFL MODELS

It follows from Propositon 2 that the set of monotonic functions in $\mathrm{KFL}(\mathcal{V})$ is fairly restricted: for example, a function in $\mathrm{KFL}(\mathcal{V})$ that is increasing in two or more directions cannot change sign in $\mathcal{D}_\mathcal{V}$. To make the model class larger, we use ensembles of sub-models in $\mathrm{KFL}(\mathcal{V})$ and take the average or the sum of their predictions as the composite model's prediction. We are thus motivated to define the class of functions that can be expressed as sums of $M$ KFL models.

$$\mathrm{KFL}(\mathcal{V}, M) = \Big\{ \sum_{i=1}^{M} g_i : g_i \in \mathrm{KFL}(\mathcal{V}) \Big\}. \tag{10}$$

Since the zero function with domain $\mathcal{D}_\mathcal{V}$ is in $\mathrm{KFL}(\mathcal{V})$, we have $\mathrm{KFL}(\mathcal{V}, M) \subseteq \mathrm{KFL}(\mathcal{V}, M+1)$. Let $\mathcal{L}(\mathcal{V})$ denote the set of all multilinear intepolation lattice functions on a grid of size $\mathcal{V}$. Then, by construction, $\mathcal{L}$ is closed under addition. So we have $\mathrm{KFL}(\mathcal{V}) = \mathrm{KFL}(\mathcal{V}, 1) \subseteq \mathrm{KFL}(\mathcal{V}, 2) \subseteq \ldots \subseteq \mathcal{L}(\mathcal{V})$. Moreover, the following proposition shows that there exists a positive integer $M$, such that for all $m < M$, there are functions that can be expressed as a sum of $m+1$ $\mathrm{KFL}(\mathcal{V})$ models which cannot be expressed as a sum of $m$ $\mathrm{KFL}(\mathcal{V})$ models, and every function in $\mathcal{L}(\mathcal{V})$ can be expressed as a sum of $M$ functions in $\mathrm{KFL}(\mathcal{V})$.

To state the proposition, we require the following definitions on *tensors*, which are multidimensional generalizations of matrices. There are multiple ways to define a real tensor. Here we view a real tensor of size $\mathcal{V}$ as a $D$-dimensional $\mathcal{V}[1] \times \ldots \times \mathcal{V}[D]$ array of real numbers. The set of such tensors is denoted by $\mathbb{R}^{\mathcal{V}[1]} \otimes \ldots \otimes \mathbb{R}^{\mathcal{V}[D]}$. We index the entries of tensors of size $\mathcal{V}$ by $\mathcal{M}_\mathcal{V}$. Addition between two tensors of the same size and multiplication of a tensor by a real scalar are defined entrywise. Observe that each element in $\mathbb{R}^{\mathcal{M}_\mathcal{V}}$ can naturally be regarded as a tensor of size $\mathcal{V}$ and vice versa. The *outer product* of $D$ vectors $\mathbf{w}_i \in \mathbb{R}^{\mathcal{V}[i]}$, $i \in [D]$ is their Kronecker product, $\otimes_{i=1}^{D} \mathbf{w}_i$, regarded as a tensor of size $\mathcal{V}$, and we use the same notation for the outer product as for the Kronecker product. A real tensor is called *simple* if it is the outer product of some $D$ real vectors. Every tensor $T$ can be expressed as a sum of simple tensors and the *rank* of $T$, denoted $r(T)$, is the minimum number of simple tensors that sum to $T$. The rank of a tensor of size $\mathcal{V}$ is at most $|\mathcal{M}_\mathcal{V}| / \max_i \{\mathcal{V}[i]\}$. See (Rabanser et al., 2017) for an introduction to tensors.

**Proposition 5.** *For two sets $A, B$ we use $A \subsetneq B$ to denote that $A$ is a subset of $B$ but $A \neq B$. Let $M = \max\{r(T) : T \in \mathbb{R}^{\mathcal{V}[1]} \otimes \ldots \otimes \mathbb{R}^{\mathcal{V}[D]}\}$. Then $\mathrm{KFL}(\mathcal{V}, 1) \subsetneq \mathrm{KFL}(\mathcal{V}, 2) \subsetneq \ldots \subsetneq \mathrm{KFL}(\mathcal{V}, M) = \mathcal{L}(\mathcal{V})$.*

See Appendix A for the proof.

## 3.4 DEPENDENCE OF THE CAPACITY OF THE MODEL CLASS ON LATTICE SIZE

In the previous section, we showed that the capacity of $\mathrm{KFL}(\mathcal{V}, M)$ increases as $M$ increases up to a certain threshold. One may also ask how the capacity of $\mathrm{KFL}(\mathcal{V}, M)$ depends on $\mathcal{V}$. Unfortunately, the domain $\mathcal{D}_\mathcal{V}$ of a member in $\mathrm{KFL}(\mathcal{V}, M)$ is different for different vectors $\mathcal{V}$, which introduces a technical difficulty when trying to compare the capacity across different lattice sizes. In practice, we follow the technique of Gupta et al. (2016) and use a "calibration" transformation $\mathbf{c} : \mathbb{R}^D \to \mathcal{D}_\mathcal{V}$ to map the "raw" input $\mathbf{x}$ into the lattice domain $\mathcal{D}_\mathcal{V}$. Here, $\mathbf{c}(\mathbf{x}) = (c_1(x[1]), \ldots, c_D(x[D]))$, and each $c_d : \mathbb{R} \to [0, \mathcal{V}[d] - 1]$ is a learned piecewise linear function with a fixed prespecified number of linear segments. See Gupta et al. (2016)) for more details. Let $\mathcal{C}(\mathcal{V})$ denote the set of all calibration transformations whose image is in $\mathcal{D}_\mathcal{V}$. For a positive integer $M$, we define the calibrated Kronecker Factored Lattice model class CKFL by: $\mathrm{CKFL}(\mathcal{V}, M) = \{f(\mathbf{c}(\cdot)) : f \in \mathrm{KFL}(\mathcal{V}, M), \mathbf{c} \in \mathcal{C}(\mathcal{V})\}$.

The following proposition shows that the capacity of $\mathrm{CKFL}(\mathcal{V}, M)$ increases as $\mathcal{V}$ increases.

**Proposition 6.** *Let $\mathcal{V}_1, \mathcal{V}_2 \in \mathbb{N}^D$ be two lattice sizes and $M$ be a positive integer. If $\mathcal{V}_1 \leq \mathcal{V}_2$, then $\mathrm{CKFL}(\mathcal{V}_1, M) \subseteq \mathrm{CKFL}(\mathcal{V}_2, M)$.*

See Appendix A for the proof.

## 3.5 MODEL TRAINING DETAILS

To ensure that the input to a lattice model lies in its domain, we follow the input calibration techniques explained in Gupta et al. (2016). For numeric inputs, we use a learned one-dimensional piecewise

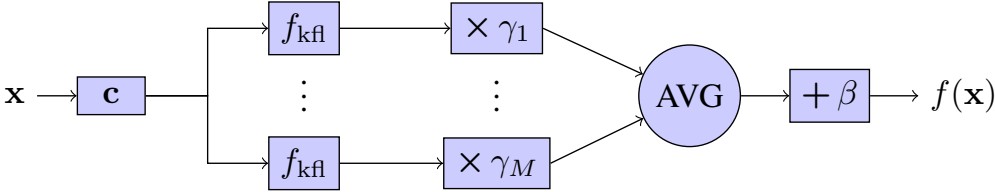

Figure 3: KFL Model Architecture

linear function to transform the feature's input domain to $[0, \mathcal{V}[d] - 1]$. For categorical inputs, we learn a one dimensional embedding for each category value to map to a real value in $[0, \mathcal{V}[d] - 1]$. Calibrators can be efficiently trained to be monotonic.

Our model architecture for KFL is depicted in Figure 3 and defined by the following equation

$$f(\mathbf{x}) = \beta + \frac{1}{M} \sum_{m \in [M]} \gamma_m \, f_{\text{kfl}}\big(\mathbf{c}(\mathbf{x}); \, \mathbf{w}_1^{(m)}, \ldots, \mathbf{w}_D^{(m)}\big),$$

where $\mathbf{c}$ is the input calibration layer, described above (note that $\mathbf{c}$ has its own learned parameters we don't show here to simplify the notation) and $\gamma_m, \beta \in \mathbb{R}$ are additional scaling and bias parameters, respectively. To guarantee that the final model is monotonic in a given feature, it is sufficient to impose monotonicity on its calibrator and each of the terms in the sum.

We train our model by minimizing the empirical risk subject to the monotonicity constraints using projected mini-batch gradient descent. Following Proposition 2, after every gradient descent step, we project each $\mathbf{w}_d^{(m)}$ to be nonnegative where required. Then for each monotonic feature $d$ we project $\mathbf{w}_d^{(m)}$ so that its components increase if $\sigma = 1$ decrease if $\sigma = -1$. Computing this projection is an instance of the Isotonic Regression problem and can be solved efficiently using the "pool-adjacent-violators" algorithm (Barlow et al., 1972). In our experiments, we instead use a more efficient approximation algorithm that computes a vector that satisfies the constraints, but may not be the closest (in $L_2$-norm) to the original vector. See Appendix B for the exact algorithm we use.

## 4 EXPERIMENTS

We present experimental results on three datasets, summarized in Table 2. The first dataset is the same public Adult Income dataset (Dheeru & Karra Taniskidou, 2017) with the same monotonicity constraint setup described in Canini et al. (2016). The other two datasets are provided by a large internet services company with monotonic features specified by product groups, as domain knowledge or policy requirements.

Table 2: Experiment Dataset Summary

| Dataset | Type | # Features | # Monotonic | # Train | # Test |
|---|---|---|---|---|---|
| Adult Income | Classification | 14 | 4 | 32,561 | 16,282 |
| Query Result Matching | Regression | 14 | 11 | 805,660 | 201,415 |
| User Query Intent | Classification | 24 | 16 | 522,302 | 130,576 |

We compare KFL against two previously proposed baseline lattice models: (1) calibrated multilinear interpolation lattice, and (2) calibrated simplex interpolation lattice, which use the input calibration layer described in Section 3.5 and Gupta et al. (2016). We note that we do not compare KFL to deep lattice networks (DLN) (You et al., 2017) in this paper. In a DLN, KFL would act as a replacement layer for the lattice layers, which only comprise a subset of the model layers. We believe that such a comparison would be dominated by the other layers and not properly show how KFL differs from previously proposed lattice models. All our models were implemented using TensorFlow (Abadi et al., 2015) and TensorFlow Lattice (Google AI Blog, 2017). Open-source code for KFL has been pushed to the TensorFlow Lattice 2.0 library and can be downloaded at github.com/tensorflow/lattice. We used logistic loss for classification and mean squared error (mse) loss for regression. For each

experiment, we train for 100 epochs with a batch size of 256 using the Adam optimizer and validate the learning rate from $\{0.001, 0.01, 0.1, 1.0\}$ with five-fold cross-validation. We use a lattice size with the same entry $V$ for each direction and tune $V$ from $\{2, 4, 8\}$ and different settings of $M$ from $[1, 100]$ for KFL (we note that increasing $V$ above 2 for our baselines was not feasible on our datasets). The results were averaged over 10 independent runs. The train and evaluation times were measured on a workstation with 6 Intel Xeon W-2135 CPUs. We report the models with best cross-validation results, as well as some other settings for comparison.

## 4.1 UCI ADULT INCOME

For the Adult Income dataset Dheeru & Karra Taniskidou (2017), the goal is to predict whether or not income is greater than or equal to $50,000. We followed the same setup described in Canini et al. (2016), specifying capital gain, weekly hours of work, education level, and the gender wage gap as monotonically increasing features. The results are summarized in Table 3 and Figure 4.

Table 3: UCI Adult Income Results

|  | $M$ | $V$ | Train Acc. | Test Acc. | Train Time (s) | Eval Time ($\mu$s) | # Parameters |
|---|---|---|---|---|---|---|---|
| Multilinear | - | 2 | $85.60 \pm 0.02$ | $85.51 \pm 0.05$ | 333 | 3700 | $16,552$ |
| Simplex | - | 2 | $86.71 \pm 0.21$ | $85.72 \pm 0.15$ | 151 | 2200 | $16,552$ |
| KFL | 1 | 4 | $86.31 \pm 0.06$ | $86.05 \pm 0.07$ | 70 | 1800 | 226 |
| KFL | 2 | 8 | $86.51 \pm 0.02$ | $86.24 \pm 0.05$ | 85 | 2100 | 395 |

## 4.2 QUERY RESULT MATCHING

This experiment is a regression problem of predicting the matching quality score of a result to a query. The results are summarized in Table 4 and Figure 5.

Table 4: Query Result Matching Results

|  | $M$ | $V$ | Train MSE | Test MSE | Train Time (s) | Eval Time ($\mu$s) | # Parameters |
|---|---|---|---|---|---|---|---|
| Multilinear | - | 2 | $0.5694 \pm 0.0010$ | $0.5702 \pm 0.0009$ | 4606 | 2835 | $16,512$ |
| Simplex | - | 2 | $0.5718 \pm 0.0066$ | $0.5732 \pm 0.0066$ | 2150 | 1952 | $16,512$ |
| KFL | 25 | 8 | $0.5728 \pm 0.0054$ | $0.5735 \pm 0.0055$ | 870 | 1847 | 2954 |
| KFL | 50 | 4 | $0.5700 \pm 0.0037$ | $0.5707 \pm 0.0038$ | 868 | 1839 | 2979 |

## 4.3 USER QUERY INTENT

For this real-world problem, the goal is to classify the user intent into one of two classes. For the baseline multilinear and simplex models, a single lattice with all 24 features is infeasible. When the number of features is too large for a single lattice, we follow the technique described in Canini et al. (2016), which uses an ensemble of lattice sub models–each taking a small subset of the calibrated input features. The result of the model is the average of the sub-models' outputs. The calibrators are shared among all lattices. Here we use a random tiny lattice ensemble of $L$ lattices with each seeing a random subset of 10 of the 24 features. We tune and set $L = 100$. The results are summarized in Table 5 and Figure 6.

Table 5: User Query Intent Results

|  | $M$ | $V$ | Train Acc. | Test Acc. | Train Time (s) | Eval Time ($\mu$s) | # Parameters |
|---|---|---|---|---|---|---|---|
| Multilinear | - | 2 | $69.48 \pm 0.02$ | $69.20 \pm 0.01$ | $13,559$ | $15,932$ | $102,619$ |
| Simplex | - | 2 | $69.48 \pm 0.03$ | $69.34 \pm 0.02$ | 4478 | 6559 | $102,619$ |
| KFL | 100 | 4 | $69.53 \pm 0.04$ | $69.35 \pm 0.04$ | 1842 | 2534 | 9920 |
| KFL | 100 | 8 | $69.74 \pm 0.08$ | $69.53 \pm 0.07$ | 3507 | 2560 | $19,520$ |

## 5 DISCUSSION

For each one of our experiments, we can see very similar results: (1) the accuracy or mse of KFL is comparable to or slightly better than the baselines, (2) the training time is significantly reduced, (3) the number of parameters is significantly reduced, (4) KFL can use larger lattice sizes $V$ because the number of parameters scales linearly and not as a power of $D$, (5) increasing $M$ increases the capacity of the KFL model class where the value for $M$ that makes KFL expressive enough to perform well is a relatively small (and going above such an $M$ no longer provides any more value), and (6) the evaluation speed of KFL compared to multilinear and simplex interpolation aligns with our theoretical runtime analysis, where KFL is significantly faster than multilinear interpolation (particularly when the number of features is large, e.g. Experiment 4.3) and comparable to simplex interpolation.

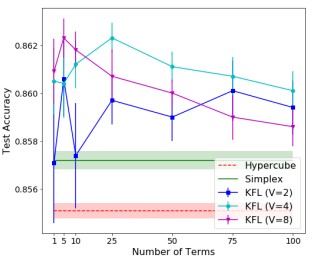 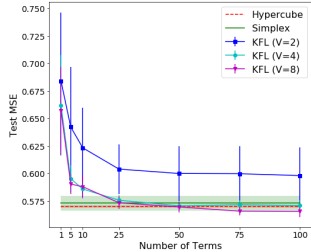 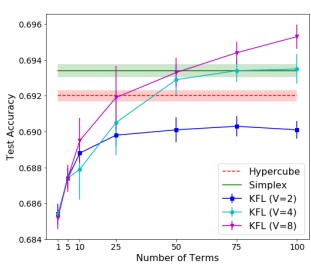

Figure 4: UCI Adult Income   Figure 5: Query Result Matching   Figure 6: User Query Intent

Figure 7: Comparison of test results of KFL with different $M$ and $V$ values

Furthermore, each experiment highlights different qualities of KFL. Experiment 4.1 shows that increasing $V$ can sometimes be more important than increasing $M$, where such an increase can potentially lead to further boosts in performance with only a linear impact to the model's time and space efficiency. This result follows from Proposition 6. In Figure 4, we see that KFL acts like a form of regularization: KFL's accuracy drops as $M$ gets too high, indicating that we start to over fit and lose this regularization effect as the size of the model class increases. Experiment 4.2 shows that KFL can also perform well for regression tasks but will likely need a larger $M$ and $V$ for more complex problems. It also shows a trade-off between the values of $M$ and $V$ where computation time, number of parameters, and mse remain pretty much fixed. Experiment 4.3 demonstrates KFL's ability to create higher-order feature interactions by keeping all features in a single lattice. Unlike multilinear and simplex interpolated lattice functions, KFL can handle many more features in a single lattice. It demonstrates that KFL can achieve even faster evaluations compared to the ensembling method described by Canini et al. (2016). Experiments 4.2 and 4.3 also show that there is a logarithmic-like improvement of performance with respect to $M$, indicating that the $M$ needed to achieve comparable accuracy is potentially impractically large (likely for more complex problems); however, as Proposition 6 shows, there is a nice trade-off between $M$ and $V$ where we can instead increase $V$ to compensate and ultimately achieve comparable or better performance with a reasonable $M$.

## 6 CONCLUSION

We proposed KFL, which reparameterizes monotonic lattice regression using Kronecker factorization to achieve significant improvement in parameter and computational efficiency. We proved sufficient and necessary conditions to efficiently impose multiple constraints during training on a model in KFL w.r.t a subset of its features. We showed that through ensembling the KFL function class strictly increases as more base KFL models are added, becoming the class of all multilinear interpolated lattice functions when the number of base models is sufficiently large. Our experimental results demonstrated its practical advantage over the multilinear and simplex interpolated lattice models in terms of speed and storage space while maintaining accuracy. For future work, it would be interesting to (1) explore applications of KFL in more complex architectures such as deep lattice networks (You et al., 2017) and (2) to understand if we can impose other useful shape constraints (Cotter et al., 2019; Gupta et al., 2018).

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

## SUPPLEMENTAL MATERIAL

## A PROOFS

**Proposition 1.** *Let $\mathcal{V} \in \mathbb{N}^D$. For all $\mathbf{w}_1 \in \mathbb{R}^{\mathcal{V}[1]}, \ldots, \mathbf{w}_D \in \mathbb{R}^{\mathcal{V}[D]}$*

$$f_{\mathrm{kfl}}(\mathbf{x}; \mathbf{w}_1, \ldots, \mathbf{w}_D) = \prod_d f_{\mathrm{pwl}}(\mathbf{x}[d]; \mathbf{w}_d), \quad \text{for all } \mathbf{x} \in \mathcal{D}_{\mathcal{V}}. \tag{5}$$

*Proof of Proposition 1.* For $v \in \mathbb{N}$, let $\mathbf{\Psi}_v : \mathbb{R} \to \mathbb{R}^v$ be given by $\mathbf{\Psi}_v(x) = [f_{\mathrm{pwl}}(x; \mathbf{e}_{1,v}) \quad f_{\mathrm{pwl}}(x; \mathbf{e}_{2,v}) \quad \ldots \quad f_{\mathrm{pwl}}(x; \mathbf{e}_{v,v})]$. Let $\mathbf{w}_1, \ldots, \mathbf{w}_D$ be vectors satisfying the conditions in the proposition. By the definition of $\mathrm{KFL}(\mathcal{V})$, we have $f_{\mathrm{kfl}}(\cdot; \mathbf{w}_1, \ldots, \mathbf{w}_D) = f_{\mathrm{mll}}(\cdot; \otimes_{d=1}^D \mathbf{w}_d)$. Fix $\mathbf{\Phi} = \mathbf{\Phi}_{\mathcal{V}}^{\mathrm{multilinear}}$. It follows from (2) that for all $\mathbf{x} \in \mathcal{D}_{\mathcal{V}}$ we have $\mathbf{\Phi}(\mathbf{x}) = \otimes_{d=1}^D \mathbf{\Psi}_{\mathcal{V}[d]}(\mathbf{x}[d])$. Therefore using (4) we get, for all $\mathbf{x} \in \mathcal{D}_{\mathcal{V}}$,

$$f_{\mathrm{kfl}}(\mathbf{x}; \mathbf{w}_1, \ldots, \mathbf{w}_D) = f_{\mathrm{mll}}(\mathbf{x}; \otimes_{d=1}^D \mathbf{w}_d) = (\otimes_{d=1}^D \mathbf{w}_d) \cdot (\otimes_{d=1}^D \mathbf{\Psi}_{\mathcal{V}[d]}(\mathbf{x}[d]))$$

$$= \prod_{d=1}^D (\mathbf{w}_d \cdot \mathbf{\Psi}_{\mathcal{V}[d]}(\mathbf{x}[d])). \tag{6}$$

Now, for each $d \in [D]$ we have

$$\mathbf{w}_d \cdot \mathbf{\Psi}_{\mathcal{V}[d]}(\mathbf{x}[d]) = \sum_{i \in [\mathcal{V}[d]]} \mathbf{w}_d[i] f_{\mathrm{pwl}}(\mathbf{x}[d]; \mathbf{e}_{i,\mathcal{V}[d]}) = f_{\mathrm{pwl}}\left(\mathbf{x}[d]; \sum_i \mathbf{w}_d[i] \mathbf{e}_{i,\mathcal{V}[d]}\right)$$

$$= f_{\mathrm{pwl}}(\mathbf{x}[d]; \mathbf{w}_d). \tag{7}$$

Substituting (7) into (6), we obtain (5). $\qquad\square$

**Proposition 2.** *Fix $\mathcal{V} \in \mathbb{N}^D$, let $f \in \mathrm{KFL}(\mathcal{V})$ and let $i_1, \ldots, i_p \in [D]$ be distinct direction indices for $p \leq D$. Then $f$ is increasing in directions $i_1, \ldots, i_p$ iff $f$ can be written as:*

$$f(\mathbf{x}) = \sigma \prod_d f_{\mathrm{pwl}}(\mathbf{x}[d]; \mathbf{w}_d), \tag{8}$$

*where $\sigma \in \{+1, -1\}$, and $\mathbf{w}_d \in \mathbb{R}^{\mathcal{V}[d]}, d \in [D]$ are vectors satisfying the following 3 conditions:*

1. *If $p \geq 2$ then $\mathbf{w}_d \geq \mathbf{0}$ for all $d \in [D]$.*

2. *If $p = 1$ then $\mathbf{w}_d \geq \mathbf{0}$ for all $d \in [D] \setminus \{i_1\}$.*

3. *For all $j \in [p]$, $\sigma \mathbf{w}_{i_j}[1] \leq \sigma \mathbf{w}_{i_j}[2] \leq \ldots \leq \sigma \mathbf{w}_{i_j}[\mathcal{V}[i_j]]$.*

To prove the proposition we shall make use of the following lemma.

**Lemma 1.** *Let $f(\mathbf{x}) = \prod_d f_{\mathrm{pwl}}(\mathbf{x}[d]; \mathbf{w}_d)$ be a function in $\mathrm{KFL}(\mathcal{V})$. If for some $i \in [D]$, $f$ is increasing in direction $i$ in $\mathcal{D}_{\mathcal{V}}$ then for all $d \in [D] \setminus \{i\}$, either $\mathbf{w}_d \geq \mathbf{0}$ or $\mathbf{w}_d \leq \mathbf{0}$.*

*Proof of Lemma 1.* Let $f$ be as above, and assume to the contrary that for some $j \in [D] \setminus \{i\}$, neither $\mathbf{w}_j \geq \mathbf{0}$ nor $\mathbf{w}_j \leq \mathbf{0}$ holds. Then there must exist $s, t \in [\mathcal{V}[j]]$ such that $\mathbf{w}_j[s] < 0$ and $\mathbf{w}_j[t] > 0$. For $d \in [D]$, let $\ell_d(x) = f_{\mathrm{pwl}}(x; \mathbf{w}_d)$. Then $\ell_j(s-1) = \mathbf{w}_j[s] < 0$ and $\ell_j(t-1) = \mathbf{w}_j[t] > 0$. By our assumption there exists $\mathbf{x} \in \mathcal{D}_{\mathcal{V}}$ and $\delta > 0$, such that $f(\mathbf{x} + \delta \mathbf{e}_i) - f(\mathbf{x}) > 0$.[1] Now, for any $r \in [0, \mathcal{V}[j] - 1]$

$$f(\mathbf{x}+(r-\mathbf{x}[j])\mathbf{e}_j+\delta\mathbf{e}_i)-f(\mathbf{x}+(r-\mathbf{x}[j])\mathbf{e}_j) = \ell_i(\mathbf{x}[i]+\delta)\ell_j(r)\prod_{d\in[D]\setminus\{i,j\}}\ell_d(\mathbf{x}[d]) - \ell_j(r)\prod_{d\in[D]\setminus\{j\}}\ell_d(\mathbf{x}[d])$$

$$= \left(\ell_i(\mathbf{x}[i]+\delta)\prod_{d\in[D]\setminus\{i\}}\ell_d(\mathbf{x}[d]) - \prod_{d\in[D]}\ell_d(\mathbf{x}[d])\right)\frac{\ell_j(r)}{\ell_j(\mathbf{x}[j])}$$

$$= (f(\mathbf{x}+\delta\mathbf{e}_i)-f(\mathbf{x}))\frac{\ell_j(r)}{\ell_j(\mathbf{x}[j])}.$$

---

[1] Otherwise $f(\mathbf{x} + \delta \mathbf{e}_i) = f(\mathbf{x})$ for all $\mathbf{x}, \delta$ implying that $f(\mathbf{x})$ does not depend on $\mathbf{x}[i]$–which we don't regard as increasing in direction $i$.

Since for some $r \in \{s-1, t-1\}$ it holds that $\ell_j(r)/\ell_j(\mathbf{x}[j]) < 0$, it follows from the last equation that for that $r$, $f(\mathbf{x}+(r-\mathbf{x}[j])\mathbf{e}_j+\delta\mathbf{e}_i) < f(\mathbf{x}+(r-\mathbf{x}[j])\mathbf{e}_j)$, which contradicts our assumption that $f$ is increasing in direction $i$. $\qquad\square$

*Proof of Proposition 2.* We first show the "if" direction. Assume that $f$ can be written as above and let $j \in [p]$. Then $f$ is the product of two terms: $\sigma f_{\mathrm{pwl}}(\mathbf{x}[i_j], \mathbf{w}_{i_j})$ and $\prod_{d \neq i_j} f_{\mathrm{pwl}}(\mathbf{x}[d]; \mathbf{w}_d)$. Condition 3, implies the first term is increasing in direction $i_j$. The second term doesn't depend on $\mathbf{x}[i_j]$ and conditions 1 and 2 imply that it is nonnegative over $\mathcal{D}_\mathcal{V}$. It follows that $f$, the product of the two terms, is increasing in direction $i_j$.

We next show the "only if" direction. To simplify the notation, we only prove this for the case $p \geq 2$. A similar argument shows the proposition holds for $p = 1$, as well. Let $f \in \mathrm{KFL}(\mathcal{V})$ be increasing in directions $i_1, \ldots, i_p$ in $\mathcal{D}_\mathcal{V}$. By Proposition 1 there exists $D$ vectors, $\mathbf{w}_d' \in \mathbb{R}^{\mathcal{V}[d]}$, $d \in [D]$, such that $f(\mathbf{x}) = \prod_d f_{\mathrm{pwl}}(\mathbf{x}[d]; \mathbf{w}_d')$. Let $\mathbf{w}_d$ be $-\mathbf{w}_d'$ if $\mathbf{w}_d' \leq \mathbf{0}$ or $\mathbf{w}_d'$, otherwise. By Lemma 1 applied to directions $i_1$ and $i_2$, it follows that for every $d \in [D]$, $\mathbf{w}_d \geq \mathbf{0}$. Since for any scalar $\alpha \in \mathbb{R}$, $f_{\mathrm{pwl}}(x[d]; \alpha\mathbf{w}_d') = \alpha f_{\mathrm{pwl}}(x[d]; \mathbf{w}_d')$, it follows that $f(\mathbf{x}) = \sigma \prod_d f_{\mathrm{pwl}}(\mathbf{x}[d]; \mathbf{w}_d)$, where $\sigma = (-1)^{|\{d : \mathbf{w}_d' \leq \mathbf{0}\}|}$. Therefore condition 1 holds.

As for condition 3, let $j \in [p]$ and assume to the contrary that $\sigma\mathbf{w}_{i_j}[s] > \sigma\mathbf{w}_{i_j}[s+1]$ for some $s \in [\mathcal{V}[i_j]-1]$. Since $f$ is not identically zero in $\mathcal{D}_\mathcal{V}$ (as we don't consider the zero function increasing in direction $i_j$), for each $d \in [D]$, there must be $x_d \in [0, \mathcal{V}[d]-1]$ such that $f_{\mathrm{pwl}}(x_d; \mathbf{w}_d) \neq 0$. Since $\mathbf{w}_d \geq \mathbf{0}$, it follows that $f_{\mathrm{pwl}}(x_d; \mathbf{w}_d) > 0$. Let $\mathbf{x} \in \mathbb{R}^D$ be the vector with $\mathbf{x}[d] = x_d$ if $d \neq i_j$ and $\mathbf{x}[i_j] = s$, otherwise. Then we have $f(\mathbf{x}) = \sigma f_{\mathrm{pwl}}(s; \mathbf{w}_{i_j}) \prod_{d \neq i_j} f_{\mathrm{pwl}}(x_d; \mathbf{w}_d) = \sigma\mathbf{w}_{i_j}[s] \prod_{d \neq i_j} f_{\mathrm{pwl}}(x_d; \mathbf{w}_d) > \sigma\mathbf{w}_{i_j}[s+1] \prod_{d \neq i_j} f_{\mathrm{pwl}}(x_d; \mathbf{w}_d) = f(\mathbf{x} + \mathbf{e}_{i_j})$ contradicting our assumption that $f$ is increasing in direction $i_j$. Therefore condition 3 holds as well. $\qquad\square$

**Proposition 3.** *Fix $\mathcal{V} \in \mathbb{N}^D$, let $f \in \mathrm{KFL}(\mathcal{V})$ and let $i_1, \ldots, i_p \in [D]$ be distinct direction indices for $p \leq D$. Then $f$ is convex in directions $i_1, \ldots, i_p$ iff $f$ can be written as:*

$$f(\mathbf{x}) = \sigma \prod_d f_{\mathrm{pwl}}(\mathbf{x}[d]; \mathbf{w}_d), \tag{9}$$

*where $\sigma \in \{+1, -1\}$, and $\mathbf{w}_d \in \mathbb{R}^{\mathcal{V}[d]}, d \in [D]$ are vectors satisfying the following 3 conditions:*

1. *If $p \geq 2$ then $\mathbf{w}_d \geq \mathbf{0}$ for all $d \in [D]$.*

2. *If $p = 1$ then $\mathbf{w}_d \geq \mathbf{0}$ for all $d \in [D] \setminus \{i_1\}$.*

3. *For all $j \in [p]$, $\mathcal{V}[i_j] = 2$ or $\mathcal{V}[i_j] > 2$ and $\sigma(\mathbf{w}_{i_j}[k+2] - \mathbf{w}_{i_j}[k+1]) \geq \sigma(\mathbf{w}_{i_j}[k+1] - \mathbf{w}_{i_j}[k])$ for all $k \in [\mathcal{V}[i_j] - 2]$.*

To prove the proposition we shall make use of the following lemma.

**Lemma 2.** *Let $f(\mathbf{x}) = \prod_d f_{\mathrm{pwl}}(\mathbf{x}[d]; \mathbf{w}_d)$ be a function in $\mathrm{KFL}(\mathcal{V})$. If for some $i \in [D]$, $f$ is convex in direction $i$ in $\mathcal{D}_\mathcal{V}$ then for all $d \in [D] \setminus \{i\}$, either $\mathbf{w}_d \geq \mathbf{0}$ or $\mathbf{w}_d \leq \mathbf{0}$.*

*Proof of Lemma 2.* Let $f$ be as above, and assume to the contrary that for some $j \in [D] \setminus \{i\}$ neither $\mathbf{w}_j \geq \mathbf{0}$ nor $\mathbf{w}_j \leq \mathbf{0}$. Then there must exist $s, q \in [\mathcal{V}[j]]$ such that $\mathbf{w}_j[s] < 0$ and $\mathbf{w}_j[q] > 0$. For $d \in [D]$, let $\ell_d(\mathbf{x}[d]) = f_{\mathrm{pwl}}(\mathbf{x}[d]; \mathbf{w}_d)$. Then $\ell_j(s-1) = \mathbf{w}_j[s] < 0$ and $\ell_j(q-1) = \mathbf{w}_j[q] > 0$. By our assumption there exists $\mathbf{x} \in \mathcal{D}_\mathcal{V}$ and $\delta > 0$ such that $tf(\mathbf{x}) + (1-t)f(\mathbf{x} + \delta\mathbf{e}_i) - f(t\mathbf{x} + (1-t)(\mathbf{x} + \delta\mathbf{e}_i)) > 0$.[2] To further simplify notation, we set $\prod_{d \neq i,j} \ell_d(\mathbf{x}[d]) = z_{d \neq i,j}$ and

---

[2]Otherwise $tf(\mathbf{x}) + (1-t)f(\mathbf{x} + \delta\mathbf{e}_i) - f(t\mathbf{x} + (1-t)(\mathbf{x} + \delta\mathbf{e}_i)) = 0$ for all $\mathbf{x}, \delta, t$ implying that $f(\mathbf{x})$ is affine in direction $i$, which we don't regard as convex in direction $i$.

$\prod_{d\neq i}\ell_d(\mathbf{x}[d]) = z_{d\neq i}$. Now, for any $r \in [0, \mathcal{V}[j]]$, we have:

$$tf(\mathbf{x} + (r - \mathbf{x}[j])\mathbf{e}_j) + (1 - t)f(\mathbf{x} + (r - \mathbf{x}[j])\mathbf{e}_j + \delta\mathbf{e}_i)$$
$$- f(t(\mathbf{x} + (r - \mathbf{x}[j])\mathbf{e}_j) + (1 - t)(\mathbf{x} + (r - \mathbf{x}[j])\mathbf{e}_j + \delta\mathbf{e}_i))$$
$$= t\ell_i(\mathbf{x}[i])\ell_j(r)z_{d\neq i,j} + (1 - t)\ell_i(\mathbf{x}[i] + \delta)\ell_j(r)z_{d\neq i,j}$$
$$- \ell_i(t\mathbf{x}[i] + (1 - t)(\mathbf{x}[i] + \delta))\ell_j(r)z_{d\neq i,j}$$
$$= \left(t\ell_i(\mathbf{x}[i])z_{d\neq i} + (1 - t)\ell_i(\mathbf{x}[i] + \delta)z_{d\neq i} - \ell_i(t\mathbf{x}[i] + (1 - t)(\mathbf{x}[i] + \delta))z_{d\neq i}\right)\frac{\ell_j(r)}{\ell_j(\mathbf{x}[j])}$$
$$= \left(tf(\mathbf{x}) + (1 - t)f(\mathbf{x} + \delta\mathbf{e}_i) - f(t\mathbf{x} + (1 - t)(\mathbf{x} + \delta\mathbf{e}_i))\right)\frac{\ell_j(r)}{\ell_j(\mathbf{x}[j])}$$

Since for some $r \in \{s - 1, q - 1\}$ it holds that $\ell_j(r)/\ell_j(\mathbf{x}[j]) < 0$, it follows from the last equation that for that $r$, $f(t(\mathbf{x} + (r - \mathbf{x}[j])\mathbf{e}_j) + (1 - t)(\mathbf{x} + (r - \mathbf{x}[j])\mathbf{e}_j + \delta\mathbf{e}_i)) > tf(\mathbf{x} + (r - \mathbf{x}[j])\mathbf{e}_j) + (1 - t)f(\mathbf{x} + (r - \mathbf{x}[j])\mathbf{e}_j + \delta\mathbf{e}_i)$, which contradicts our assumption that $f$ is convex in direction $i$ (for the point $\mathbf{x} + (r - \mathbf{x}[j])\mathbf{e}_j$). □

*Proof of Proposition 3.* We first show the "if" direction. Assume that $f$ can be written as above and let $j \in [p]$. Then $f$ is the product of two terms: $\sigma f_{\mathrm{pwl}}(\mathbf{x}[i_j]; \mathbf{w}_{i_j})$ and $\prod_{d\neq i_j} f_{\mathrm{pwl}}(\mathbf{x}[d]; \mathbf{w}_d)$. Condition 3 implies the first term is convex in direction $i_j$. The second term does not depend on $\mathbf{x}[i_j]$ and conditions 1 and 2 imply that it is non-negative over $\mathcal{D}_\mathcal{V}$. It follows that $f$, the product of the two terms, is convex in direction $i_j$.

We next show the "only if" direction. To simplify the notation, we only prove this for the case $p \geq 2$. A similar argument shows the proposition holds for $p = 1$, as well. Let $f \in \mathrm{KFL}(\mathcal{V})$ be convex in directions $i_1, ..., i_p$ in $\mathcal{D}_\mathcal{V}$. By Proposition 1 there exists $D$ vectors, $\mathbf{w}'_d \in \mathbb{R}^{\mathcal{V}[d]}$, $d \in [D]$, such that $f(\mathbf{x}) = \prod_d f_{\mathrm{pwl}}(\mathbf{x}[d]; \mathbf{w}'_d)$. Let $\mathbf{w}_d$ be $-\mathbf{w}'_d$ if $\mathbf{w}'_d \leq \mathbf{0}$ or $\mathbf{w}'_d$, otherwise. By Lemma 2 applied to directions $i_1$ and $i_2$, it follows that for every $d \in [D]$, $\mathbf{w}_d \geq \mathbf{0}$. Since for any scalar $\alpha \in \mathbb{R}$, $f_{\mathrm{pwl}}(x[d]; \alpha\mathbf{w}'_d) = \alpha f_{\mathrm{pwl}}(x[d]; \mathbf{w}'_d)$, it follows that $f(\mathbf{x}) = \sigma \prod_d f_{\mathrm{pwl}}(\mathbf{x}[d]; \mathbf{w}_d)$, where $\sigma = (-1)^{|\{d : \mathbf{w}'_d \leq \mathbf{0}\}|}$. Therefore condition 1 holds.

As for condition 3, let $j \in [p]$ and assume to the contrary that $\mathcal{V}[i_j] > 2$ but $\sigma(\mathbf{w}_{i_j}[k + 2] - \mathbf{w}_{i_j}[k + 1]) < \sigma(\mathbf{w}_{i_j}[k + 1] - \mathbf{w}_{i_j}[k])$ for some $k \in [\mathcal{V}[i_j - 2]]$. Since $f$ is not identically zero in $\mathcal{D}_\mathcal{V}$ (as we don't consider the zero function convex in direction $i_j$), for each $d \in [D]$, there must be $x_d \in [0, \mathcal{V}[d] - 1]$ such that $f_{\mathrm{pwl}}(x_d; \mathbf{w}_d) \neq 0$. Since $\mathbf{w}_d \geq \mathbf{0}$ it follows that $f_{\mathrm{pwl}}(x_d; \mathbf{w}_d) > 0$. Now let $t = 0.5$ and consider ordered inputs $\mathbf{x}_1, \mathbf{x}_2 \in \mathbb{R}^D$ such that $\mathbf{x}_1[d] = \mathbf{x}_2[d] = x_d$ for all $d \in [D]$, $\mathbf{x}_1[i_j] = k - 1$, and $\mathbf{x}_2[i_j] = k + 1$. We first decompose $f(\mathbf{x}) = \sigma f_{\mathrm{pwl}}(\mathbf{x}[i_j]; \mathbf{w}_{i_j})\prod_{d\neq i_j} f_{\mathrm{pwl}}(\mathbf{x}[d]; \mathbf{w}_d)$. Next, by construction, we note that $\prod_{d\neq i_j} f_{\mathrm{pwl}}(\mathbf{x}_1[d]; \mathbf{w}_d) = \prod_{d\neq i_j} f_{\mathrm{pwl}}(\mathbf{x}_2[d]; \mathbf{w}_d) = \prod_{d\neq i_j} f_{\mathrm{pwl}}(\mathbf{x}_3[d]; \mathbf{w}_d) = z > 0$, and since $f$ is convex in direction $i_j$, we have:

$$\sigma z f_{\mathrm{pwl}}(tx_1[i_j] + (1 - t)(x_2[i_j]); \mathbf{w}_{i_j}) \leq \sigma z t f_{\mathrm{pwl}}(x_1[i_j]; \mathbf{w}_{i_j}) + \sigma z(1 - t)f_{\mathrm{pwl}}(x_2[i_j]; \mathbf{w}_{i_j})$$
$$\sigma f_{\mathrm{pwl}}(0.5(k - 1) + 0.5(k + 1); \mathbf{w}_{i_j}) \leq 0.5\left(\sigma f_{\mathrm{pwl}}(k - 1; \mathbf{w}_{i_j}) + \sigma f_{\mathrm{pwl}}(k + 1; \mathbf{w}_{i_j})\right)$$
$$2\sigma\mathbf{w}_{i_j}[k + 1] \leq \sigma\mathbf{w}_{i_j}[k] + \sigma\mathbf{w}_{i_j}[k + 2]$$
$$\sigma(\mathbf{w}_{i_j}[k + 1] - \mathbf{w}_{i_j}[k]) \leq \sigma(\mathbf{w}_{i_j}[k + 2] - \mathbf{w}_{i_j}[k + 1])$$

which contradicts our assumption. □

**Proposition 4.** *Fix $\mathcal{V} \in \mathbb{N}^D$, let $f \in \mathrm{KFL}(\mathcal{V})$. Then $f$ is nonnegative (resp. strictly positive) iff $f$ can be written as $f(\mathbf{x}) = \prod_d f_{\mathrm{pwl}}(\mathbf{x}[d]; \mathbf{w}_d)$, where $\mathbf{w}_d \geq \mathbf{0}$ (resp. $\mathbf{w}_d > \mathbf{0}$) for all $d \in [D]$.*

*Proof of Proposition 4 (nonnegative case).* We first show the "if" direction. Assume that $f$ can be written as above. Then for each $d \in [D]$, since $\mathbf{w}_d \geq \mathbf{0}$, it follows that $f_{\mathrm{pwl}}(\mathbf{x}[d]; \mathbf{w}_d) \geq 0$. Therefore, $f$, the product of $D$ nonnegative piecewise linear functions, is nonnegative.

We next show the "only if" direction. By Proposition 1 there exists $D$ vectors, $\mathbf{w}'_d \in \mathbb{R}^{\mathcal{V}[d]}$, $d \in [D]$, such that $f(\mathbf{x}) = \prod_d f_{\mathrm{pwl}}(\mathbf{x}[d]; \mathbf{w}'_d)$. If for some $i \in [D]$, neither $\mathbf{w}'_i \geq \mathbf{0}$ nor $\mathbf{w}'_i \leq \mathbf{0}$ then there must exist $s, t \in [0, \mathcal{V}[i] - 1]$ such that $f_{\mathrm{pwl}}(s; \mathbf{w}'_i) < 0$ and $f_{\mathrm{pwl}}(t; \mathbf{w}'_i) > 0$.

It follows that there exists $\mathbf{x} \in \mathcal{D}_\mathcal{V}$ such that either $f_{\mathrm{pwl}}(s; \mathbf{w}'_i) \prod_{d \neq i} f_{\mathrm{pwl}}(\mathbf{x}[d]; \mathbf{w}'_d) < 0$ or $f_{\mathrm{pwl}}(t; \mathbf{w}'_i) \prod_{d \neq i} f_{\mathrm{pwl}}(\mathbf{x}[d]; \mathbf{w}'_d) < 0$, thus making $f$ not nonnegative[3]. Hence, for all $d \in [D]$, either $\mathbf{w}'_d \geq \mathbf{0}$ or $\mathbf{w}'_d \leq \mathbf{0}$. Now, let $\mathbf{w}_d$ be $-\mathbf{w}'_d$ if $\mathbf{w}'_d \leq \mathbf{0}$ or $\mathbf{w}'_d$, otherwise. Since for any scalar $\alpha \in \mathbb{R}$, $f_{\mathrm{pwl}}(\mathbf{x}[d]; \alpha \mathbf{w}_d) = \alpha f_{\mathrm{pwl}}(\mathbf{x}[d]; \mathbf{w}_d)$, it follows that $f(\mathbf{x}) = \sigma \prod_d f_{\mathrm{pwl}}(\mathbf{x}[d]; \mathbf{w}_d)$, where $\sigma = (-1)^{|\{d: \mathbf{w}'_d \leq \mathbf{0}\}|}$. Since $\prod_d f_{\mathrm{pwl}}(\mathbf{x}[d]; \mathbf{w}_d) \geq 0$, for $f$ to be nonnegative, $|\{d: \mathbf{w}'_d \leq \mathbf{0}\}|$ must be even. This implies $\sigma = 1$. Then a nonnegative function $f$ is written as $f(\mathbf{x}) = \prod_d f_{\mathrm{pwl}}(\mathbf{x}[d]; \mathbf{w}_d)$, where $\mathbf{w}_d \geq \mathbf{0}$ for all $d \in [D]$. $\square$

*Proof of Proposition 4 (strictly positive case).* We first show the "if" direction. Assume that $f$ can be written as above. Then for each $d \in [D]$, since $\mathbf{w}_d > \mathbf{0}$, it follows that $f_{\mathrm{pwl}}(\mathbf{x}[d]; \mathbf{w}_d) > 0$. Therefore, $f$, the product of $D$ strictly positive piecewise linear functions, is strictly positive.

We next show the "only if" direction. By Proposition 1 there exists $D$ vectors, $\mathbf{w}'_d \in \mathbb{R}^{\mathcal{V}[d]}$, $d \in [D]$, such that $f(\mathbf{x}) = \prod_d f_{\mathrm{pwl}}(\mathbf{x}[d]; \mathbf{w}'_d)$. If for some $i \in [D]$, neither $\mathbf{w}'_i > \mathbf{0}$ nor $\mathbf{w}'_i < \mathbf{0}$ then there must exist $\mathbf{x}[i]$ such that $f_{\mathrm{pwl}}(\mathbf{x}[i]; \mathbf{w}'_i) = 0$. Thus, it follows that $f$ is not strictly positive. Hence, for all $d \in [D]$, either $\mathbf{w}'_d > \mathbf{0}$ or $\mathbf{w}'_d < \mathbf{0}$. Now, let $\mathbf{w}_d$ be $-\mathbf{w}'_d$ if $\mathbf{w}'_d < \mathbf{0}$ or $\mathbf{w}'_d$, otherwise. Since for any scalar $\alpha \in \mathbb{R}$, $f_{\mathrm{pwl}}(\mathbf{x}[d]; \alpha \mathbf{w}_d) = \alpha f_{\mathrm{pwl}}(\mathbf{x}[d]; \mathbf{w}_d)$, it follows that $f(\mathbf{x}) = \sigma \prod_d f_{\mathrm{pwl}}(\mathbf{x}[d]; \mathbf{w}_d)$, where $\sigma = (-1)^{|\{d: \mathbf{w}'_d < \mathbf{0}\}|}$. Since $\prod_d f_{\mathrm{pwl}}(\mathbf{x}[d]; \mathbf{w}_d) > 0$, for $f$ to be strictly positive, $|\{d: \mathbf{w}'_d < \mathbf{0}\}|$ must be even. This implies $\sigma = 1$. Then a strictly positive function $f$ is written as $f(\mathbf{x}) = \prod_d f_{\mathrm{pwl}}(\mathbf{x}[d]; \mathbf{w}_d)$, where $\mathbf{w}_d > \mathbf{0}$ for all $d \in [D]$. $\square$

**Proposition 5.** *For two sets $A, B$ we use $A \subsetneq B$ to denote that $A$ is a subset of $B$ but $A \neq B$. Let $M = \max\{r(T) : T \in \mathbb{R}^{\mathcal{V}[1]} \otimes \ldots \otimes \mathbb{R}^{\mathcal{V}[D]}\}$. Then $\mathrm{KFL}(\mathcal{V}, 1) \subsetneq \mathrm{KFL}(\mathcal{V}, 2) \subsetneq \ldots \subsetneq \mathrm{KFL}(\mathcal{V}, M) = \mathcal{L}(\mathcal{V})$.*

*Proof.* We first show $\mathrm{KFL}(\mathcal{V}, M) = \mathcal{L}(\mathcal{V})$. Since $\mathcal{L}(\mathcal{V})$ is closed under addition, $\mathrm{KFL}(\mathcal{V}, M) \subseteq \mathcal{L}(\mathcal{V})$, so it suffices to show that $\mathcal{L}(\mathcal{V}) \subseteq \mathrm{KFL}(\mathcal{V}, M)$. For $\boldsymbol{\theta} \in \mathbb{R}^{\mathcal{M}_\mathcal{V}}$, denote by $f_{\mathrm{mll}}(\cdot; \boldsymbol{\theta})$ the multilinear interpolation lattice function with parameters $\boldsymbol{\theta}$. Let $f_{\mathrm{mll}}(\cdot; \boldsymbol{\theta}) \in \mathcal{L}(\mathcal{V})$. Regard $\boldsymbol{\theta}$ as a tensor of size $\mathcal{V}$ and set $r = r(\boldsymbol{\theta})$. Thus, there exist $rD$ vectors $\mathbf{w}_j^{(i)} \in \mathbb{R}^{\mathcal{V}[j]}$, $i \in [r]$, $j \in [D]$, such that $\sum_{i \in [r]} \otimes_{j=1}^D \mathbf{w}_j^{(i)} = \boldsymbol{\theta}$. Thus, from (1), it follows that $f_{\mathrm{mll}}(\cdot; \boldsymbol{\theta}) = f_{\mathrm{mll}}(\cdot; \sum_{i \in [r]} \otimes_{j=1}^D \mathbf{w}_j^{(i)}) = \sum_{i \in [r]} f_{\mathrm{mll}}(\cdot; \otimes_{j=1}^D \mathbf{w}_j^{(i)})$. Now, by the definition of $\mathrm{KFL}(\mathcal{V})$, for each $i \in [r]$, $f_{\mathrm{mll}}(\cdot; \otimes_{j=1}^D \mathbf{w}_j^{(i)}) \in \mathrm{KFL}(\mathcal{V})$, and by the definition of $M$ in the proposition it holds that $r \leq M$. Since the zero function is in $\mathrm{KFL}(\mathcal{V})$, it follows that $f_{\mathrm{mll}}(\cdot; \boldsymbol{\theta}) = \sum_{i \in [r]} f_{\mathrm{mll}}(\cdot; \otimes_{j=1}^D \mathbf{w}_j^{(i)}) \in \mathrm{KFL}(\mathcal{V}, M)$.

It remains to show that for each $m \in [M-1]$, $\mathrm{KFL}(\mathcal{V}, m) \subsetneq \mathrm{KFL}(\mathcal{V}, m+1)$. Since the zero function is in $\mathrm{KFL}(\mathcal{V})$, $\mathrm{KFL}(\mathcal{V}, m) \subseteq \mathrm{KFL}(\mathcal{V}, m+1)$. Assume to the contrary that $\mathrm{KFL}(\mathcal{V}, m+1) = \mathrm{KFL}(\mathcal{V}, m)$, and let $\boldsymbol{\theta} \in \mathbb{R}^{\mathcal{M}_\mathcal{V}}$ be a tensor of size $\mathcal{V}$ and rank $M$. Then

$$\boldsymbol{\theta} = \sum_{i \in [M]} \otimes_{j=1}^D \mathbf{w}_j^{(i)}, \tag{11}$$

for some $MD$ vectors $\mathbf{w}_j^{(i)} \in \mathbb{R}^{\mathcal{V}[j]}$, with $j \in [D]$ and $i \in [M]$. From the definition of $\mathrm{KFL}(\mathcal{V}, m+1)$, it follows that $\sum_{i \in [m+1]} f_{\mathrm{mll}}(\cdot; \otimes_{j=1}^D \mathbf{w}_j^{(i)}) \in \mathrm{KFL}(\mathcal{V}, m+1)$, thus by our assumption it is also in $\mathrm{KFL}(\mathcal{V}, m)$. Therefore there exist $mD$ vectors $\mathbf{u}_j^{(i)} \in \mathbb{R}^{\mathcal{V}[j]}$, with $j \in [m]$ and $i \in [D]$, such that $\sum_{i \in [m+1]} f_{\mathrm{mll}}(\cdot; \otimes_{j=1}^D \mathbf{w}_j^{(i)}) = \sum_{i \in [m]} f_{\mathrm{mll}}(\cdot; \otimes_{j=1}^D \mathbf{u}_j^{(i)})$, or equivalently that $f_{\mathrm{mll}}(\cdot; \sum_{i \in [m+1]} \otimes_{j=1}^D \mathbf{w}_j^{(i)}) = f_{\mathrm{mll}}(\cdot; \sum_{i \in [m]} \otimes_{j=1}^D \mathbf{u}_j^{(i)})$. Since the mapping $\boldsymbol{\theta} \to f_{\mathrm{mll}}(\cdot; \boldsymbol{\theta})$ is one-to-one (as for all $\mathbf{v} \in \mathcal{M}_\mathcal{V}$, $f(\mathbf{v}) = \theta_\mathbf{v}$, where $\theta_\mathbf{v}$ is the entry of $\boldsymbol{\theta}$ with index $\mathbf{v}$), it follows from the last equality that $\sum_{i \in [m+1]} \otimes_{j=1}^D \mathbf{w}_j^{(i)} = \sum_{i \in [m]} \otimes_{j=1}^D \mathbf{u}_j^{(i)}$. Therefore we may replace the sum of the first $m + 1$ elements in (11) with the sum of the $m$ simple tensors $\sum_{i \in [m]} \otimes_{j=1}^D \mathbf{u}_j^{(i)}$, obtaining a sum of $M - 1$ simple tensors that yields $\boldsymbol{\theta}$, contradicting the assumption that $r(\boldsymbol{\theta}) = M$. $\square$

**Proposition 6.** *Let $\mathcal{V}_1, \mathcal{V}_2 \in \mathbb{N}^D$ be two lattice sizes and $M$ be a positive integer. If $\mathcal{V}_1 \leq \mathcal{V}_2$, then $\mathrm{CKFL}(\mathcal{V}_1, M) \subseteq \mathrm{CKFL}(\mathcal{V}_2, M)$.*

---

[3]If for some $j \in [D] \setminus \{i\}$ $\mathbf{w}'_j = \mathbf{0}$ then $f$ is still nonnegative, but for this edge case, $f$ can be trivially written as $f(\mathbf{x}) = \prod_d f_{\mathrm{pwl}}(\mathbf{x}[d]; \mathbf{w}_d)$ where $\mathbf{w}_d = \mathbf{0}$.

*Proof.* Assume $\mathcal{V}_1 \leq \mathcal{V}_2$ and let $g \in \text{CKFL}(\mathcal{V}_1, M)$. We need to show that $g \in \text{CKFL}(\mathcal{V}_2, M)$, as well. By our assumption $g = f(\mathbf{c}(\cdot))$ for some $f \in \text{KFL}(\mathcal{V}_1, M)$ and $\mathbf{c} \in \mathcal{C}(\mathcal{V}_1)$. Thus, there exist vectors $\mathbf{w}_d^{(m)}$ for $m \in [M]$, $d \in [D]$, such that $f = \sum_m f_{\text{kfl}}(\cdot; \mathbf{w}_1^{(m)}, \ldots, \mathbf{w}_D^{(m)})$. For each such $m$ and $d$, define $\mathbf{w}'_d{}^{(m)} \in \mathbb{R}^{\mathcal{V}_2[d]}$ by $\mathbf{w}'_d{}^{(m)}[j] = \mathbf{w}_d^{(m)}[j]$ for $j \in [\mathcal{V}_1[d]]$, and $\mathbf{w}'_d{}^{(m)}[j] = 0$, for $j \in [\mathcal{V}_2[d]] \setminus [\mathcal{V}_1[d]]$. Let $f^* \in \text{KFL}(\mathcal{V}_2, M)$ be the function given by $f^* = \sum_m f_{\text{kfl}}(\cdot; \mathbf{w}'_1{}^{(m)}, \ldots, \mathbf{w}'_D{}^{(m)})$. Since $\mathcal{V}_1 \leq \mathcal{V}_2$, we have $\mathcal{D}_{\mathcal{V}_1} \subseteq \mathcal{D}_{\mathcal{V}_2}$, and thus $\mathbf{c} \in \mathcal{C}(\mathcal{V}_2)$, as well. Therefore, by the definition of CKFL, we have $f^*(\mathbf{c}(\cdot)) \in \text{CKFL}(\mathcal{V}_2, M)$. Now, for each $m \in [M]$, it follows from (5) and the construction of the vectors $\mathbf{w}'_d{}^{(m)}$ that for all $\mathbf{y} \in \mathcal{D}_{\mathcal{V}_1}$, we have $f_{\text{kfl}}(\mathbf{y}; \mathbf{w}'_1{}^{(m)}, \ldots, \mathbf{w}'_D{}^{(m)}) = f_{\text{kfl}}(\mathbf{y}; \mathbf{w}_1^{(m)}, \ldots, \mathbf{w}_D^{(m)})$. Consequently, for all $\mathbf{y} \in \mathcal{D}_{\mathcal{V}_1}$, $f(\mathbf{y}) = f^*(\mathbf{y})$. Since for all $\mathbf{x} \in \mathbb{R}^D$, $\mathbf{c}(\mathbf{x}) \in \mathcal{D}_{\mathcal{V}_1}$, we get that $g = f(\mathbf{c}(\cdot)) = f^*(\mathbf{c}(\cdot))$. Therefore $g \in \text{CKFL}(\mathcal{V}_2, M)$. $\qquad\square$

## B APPROXIMATION ALGORITHM FOR MONOTONICITY PROJECTION

We use Algorithm 1 to map a vector $\mathbf{w} \in \mathbb{R}^D$ to a close (in $L_2$ norm) vector $\mathbf{w}'$ that satisfies $\mathbf{w}'[1] \leq \ldots \leq \mathbf{w}'[D]$. The vector $\mathbf{w}'$ may not be the closest such vector to $\mathbf{w}$, but in our experiments the algorithm worked well enough as the projection step in the projected SGD optimization algorithm and is faster than the pool-adjacent-violators algorithm. We use an analogous algorithm to map $\mathbf{w}$ to a close vector with monotonic decreasing components. In the description of the algorithm we denote by $|\mathbf{v}|$ the $L_2$ norm of a real vector $\mathbf{v}$.

---

**input** : A vector $\mathbf{w} \in \mathbb{R}^D$
**output**: A vector $\mathbf{w}' \in \mathbb{R}^D$ s.t. $\mathbf{w}'[1] \leq \ldots \leq \mathbf{w}[D]$ and $|\mathbf{w}' - \mathbf{w}|$ is "small".

```
1  v ← w;
2  // Make v satisfy the constraints by increasing components.
3  for i ← 2 to D do
4  |   v[i] ← max{v[i], v[i−1]};
5  end
6  w' ← (v+w)/2;
7  // Make w' satisfy the constraints by decreasing components.
8  for i ← D−1 to 1 do
9  |   w'[i] = min{w'[i], w'[i+1]};
10 end
```

---

**Algorithm 1:** Approximation Algorithm for solving Simply Ordered Uniformally Weighted Isotonic Regression

