# OpenReview forum: "Monotonic Kronecker-Factored Lattice"
_ICLR.cc/2021/Conference — ICLR 2021 Poster_

### Official Review · AnonReviewer1 · 2020-10-27
**Review Comment #1**

**Rating:** 6
**Confidence:** 2

**Review:**

(Added on 11/29/2020) **Post Rebuttal Comment**

I thank the authors for sincerely replying my review comments. I think the authors answered my questions.

Addtional Comments

- Section 3.4: $(c_1(x[1]),...,c_D(x[D])$ →$(c_1(x[1]),...,c_D(x[D]))$ (Add a right parenthesis)

---------------------------------------------------------

**Review Summary**

Theoretical claims about the condition about single KFL's monotonicity and the expressive power of the ensemble of KFL's are correct. Also, the ensemble of KFL demonstrates empirical performance comparable with existing methods with fewer computational complexities up to 24-dimensional feature vectors. In the experiments, this paper learns the monotonic function using the ensemble of KFL, whose weak learners are monotonic. However, I wonder whether it is theoretically justified (see "Soundness of the claims" section).

**Summary of the Paper**

This paper proposed Kronecker-Factored Lattice (KFL) for learning monotonic functions, which is computationally efficient than the existing method in terms of input dimension. This paper derived a necessary and sufficient condition that KFL is monotonic, which is easy to check. This paper also proposed an ensemble of KFL and showed that its expressive power is sufficiently strong that any lattice-point-interpolating function when the number of base learners is sufficiently large. This paper applied the ensemble of KFL for learning monotonic function using three datasets and confirmed that the proposed methods performed comparably with the existing methods with few computational resources and time.

**Claim**

If I understand correctly, the main claims of this paper are as follows. I assume them and evaluate the paper based on whether it supports them.
- Claim 1: The computational and storage cost of the proposed method is efficient. (Contribution 1, 4)
- Claim 2: We can use KFL to learn monotonic function (Contribution 2)
- Claim 3: The ensemble of KFL is sufficiently expressive (Contribution 3)
- Claim 4: The proposed method empirically performs well (Contribution 4)

**Soundness of the claims**

Can theory support the claim?
- Claim 1 is supported by the discussion in the last paragraph of Section 2. The proposed method is $O(D)$ evaluation time and $O(\sum_d \mathcal{V}[d])$  parameters, which is typically linear in $D$.
- Regarding Claim 2, the author proposed a training method (Section 3.3, Paragraph 3) of KFL that is justified by Proposition 2.
- Claim 3 is supported by Proposition 3. This paper showed that the expressive power of the ensemble of $T$ KFL is strictly increasing in terms of $T$ up to some $T_0$. This paper also showed that the $T$ ensemble of KFL can represent any lattice-point-interpolating functino for any $T\geq T_0$.
- In the experiments, this paper learned monotonic functions using the ensemble of KFL by imposing monotonicity to base learners (i.e., KFL). However, if I understand correctly, the expressive power of the ensemble of monotonic KFL was not studied. It is true that any function in $\mathcal{L}(\mathcal{V})$ can represent the sum of KFLs, which are not necessarily monotonic. However, it is not known we can express the monotonic function using monotonic KFL's only.

Can empirical evaluation support the claim?
- Claim 1 is supported by the fact that the training time in Tables 3, 4, and 5 is reduced. Regarding storage cost, the number of parameters is reduced.
- Claim 2, 3 do not have corresponding empirical support due to their theoretical nature.
- Claim 4 is supported by the test accuracy in Tables 3, 4, and 5. At least we can observe clear improvement from multilinear and simplex methods by Gupta et al. (2016) when appropriately configuring  and $M$. However, from Figure 4--6, the ensemble of KFL does not perform well when $V$ and $M$ is small.

**Significance and novelty**

Relation to previous work (What is different from previous work)?
- Gupta et al. (2016) also proposed a method for learning monotonic function using a function that interpolates lattice points linearly. They discussed the difference between the proposed method and Gupta et al. (2016). Specifically, Gupta et al. (2016) takes $O(2^D)$ time to evaluate a $D$-variate function and has $O(\prod_d \mathcal{V}[d])$ parameters, which is typically exponential in $D$. On the other hand, those of the proposed method are typically linear (Claim 1).

Novelty
- The idea of the proposed method is reasonable to me in that we employ Kronecker Factorization to reduce the computational and storage complexities and compensate for the reduced expressive power by ensembling. The idea of ensembling is also found in Canini et al. (2016); however, this paper's strength is that it backbones this idea by theoretical results (Proposition 2, 3), which are novel.

**Correctness and Clarity**

Is the theorem correct?
- The proofs of propositions are correct, so far as I check.

Is the experimental evaluation correct?
- I do not find any inappropriate points in experimental settings.

Is the experiment reproducible?
- The implementation of the proposed method is made public. This paper used three datasets in experiments. One dataset (Adult Income) is made public, and the other two datasets are proprietary. So, it is hard for us to reproduce the same experiments using private datasets.

**Clarity**

Can I understand the main point of the paper easily?
- Yes, I can understand the backgrounds of the proposed method explained in Section 2. The mathematical descriptions of this paper are clear.

Is the organization of paper well?
- Yes. I did not find any problem with the organization of the paper.

Are figures and tables appropriately made?
- Yes

**Additional feedback**

- Section 4, Paragraph 1: This paper says that for each monotonic feature $d$, $w^{(m)}_d$ is projected to satisfy the monotonicity constraints of Proposition 2 Condition 3. I think this projection operation is not obvious per se. So, I want this paper to clarify it.
- Proposition 3: Is there a case in which $M$ is strictly smaller than the upepr bound $|\mathcal{M}_\mathcal{V}| /  \max_i \mathcal{V}_i$ ? It is merely my intuition but since $T$ can take an arbitrary tensor of size $\mathcal{V}$, there might not be such a case.

---

> ### Author Response · Authors · 2020-11-17
> **Response**
>
> We would like to thank the reviewer for their thoughtful feedback!
>
> 1. It may well be that some monotonic functions in L(V) cannot be expressed as a sum of monotonic KFL functions. In fact, the same caveat may also be true for the technique of constraining monotonicity of ensembles of conventional multilinear lattices described in Canini et al. (2016), where each base lattice is constrained to be monotonic. In practice, though, we often find that ensembles are expressive enough for most applications, which is further demonstrated by our empirical results. Thank you very much for pointing this out.
>
> 2. We updated the paper to show the exact algorithm used (see Section 3.5 and Appendix B). Thank you for pointing this out.
>
> 3. Thank you for the good question! No, unfortunately, the bound is not tight. For example, for $2 \times 2 \times 2$ real tensors, the maximum rank (over the real field) is 3 not 4 (see https://arxiv.org/pdf/math/0607647.pdf, Section 7), and for general n x n x n real tensors, the maximum rank (again, over the real field) is at most $n(n+1)/2$ (see https://link.springer.com/article/10.1007/s10463-010-0294-5 , Theorems 1, and 3). As far as we know, the problem of finding the maximum rank for a general tensor size is open.

---

> > ### Comment · AnonReviewer1 · 2020-11-20
> > **Reply to authors' response**
> >
> > I thank the authors to consider my review comment seriously and answer my questions. I understand the authors' questions. Let me take time to check the updated version of the paper. I will ask additional questions if I have them.

---

### Official Review · AnonReviewer3 · 2020-10-28
**Novel reparametrization of monotonic lattice regression which shows significant empirical gains.**

**Rating:** 7
**Confidence:** 3

**Review:**

**Summary**
The authors propose KFL, an efficient reparametrization of monotonic lattice regression using Kronecker factorization. The goal is to achieve efficiency both in terms of computations and in terms of the number of parameters. The authors show that the proposed KFL has storage and computational costs that scale linearly in the number of input features.
Experimental results show that KFL has better speed and storage space. The authors also provide necessary and sufficient conditions for a KFL model to be monotonic with respect to some features.

**+ves**
+ This appears to be a new parametrization of monotonic lattice regression with parametric and computational advanages
+ Theoretical results show necessary and sufficient conditions for KFL to be monotonic, and this allows the design of training algorithms; there are also theoretical results on the capacity
+ Experiments on public and proprietary datasets show that KFL maintains error rate while needing less time to train and fewer parameters.

**Concerns**
- The gains in training and eval time (especially compared to simplex) seem modest. However the number of parameters is significantly reduced. I wonder if a larger dataset/more complex task would demonstrate the benefits more clearly.

---

> ### Author Response · Authors · 2020-11-17
> **Response**
>
> We thank the reviewer for their thoughtful feedback!
>
> In regards to the train/eval time comparison against Simplex, we would like to first reiterate that while Simplex’s space complexity is $O(V^D)$, the time complexity to evaluate one example is $O(Dlog(D))$. This is because Simplex partitions the unit hypercube into $D!$ simplices (for $V=2$), and finds the simplex containing the example during inference. Hence, for the first two  datasets with 14 features, the evaluation times are roughly similar for KFL and Simplex. However, during training, examples in a mini-batch do not necessarily have the same containing simplex, and training thus requires access to many simplices resulting in expensive matrix operations for each gradient update step. We show KFL is up to $2.5x$ faster to train than Simplex and, as mentioned by the reviewer, we also expect the training time gap to widen as the number of features grows. In fact, for the User Query Intent dataset, a single Simplex lattice with all 24 features takes ~2hrs to train for a single epoch. We estimate this to be $400x$ slower than KFL with $M=100$. In the reported experiment, however, Simplex used a random tiny lattice ensemble of 100 lattices with each seeing a random subset of 24 features as described in Canini et al. (2016) instead to achieve efficient training. This restricts the model capacity as it cannot capture full non-linear feature interaction, so our KFL model with $M=100$, $V=8$ is shown to be significantly more accurate, yet faster and more compact.

---

### Official Review · AnonReviewer2 · 2020-10-29
**Faster model with the factorization**

**Rating:** 6
**Confidence:** 1

**Review:**

The paper proposes to use Kronecker factorization on a lattice for a monotonic lattice. Thanks to the factorization, the computational cost is improved to linear compared to exponential. The paper went on to extend to use an ensemble of KFLs due to each factorized lattice is too restricted. Compared with the baseline, monotonic KFL achieves good accuracy with much faster run times on the experiments.

The approach seems to strike a good balance between computation and representation of power by choosing an ensemble of factorized functions.

I am not familiar with the motivation for monotonic functions and why the datasets/tasks such as query result matching requires the resulting function to satisfy monotonicity. For this reason, I am not sure about the significance of the work.

---

> ### Author Response · Authors · 2020-11-17
> **Response**
>
> We would like to thank the reviewer for their thoughtful comments and address the motivations behind monotonic functions and their significance.
>
> Consider a bank that wants to create a model that decides whether or not to approve a loan for a client where one of the features is the client’s credit score. It follows that if all other features are frozen, increasing credit score should always correspond to an increased chance of approval; however, an unconstrained model may learn a function that does not properly learn this relationship. This is an example of what makes monotonicity constraints significant -- we can impose this real-world knowledge constraint on the model from the beginning, which often results in better generalization, interpretability, and confidence that the model will perform as expected especially when the distribution of data used during inference differs from the training data distribution. In this loan approval case, it may actually be against regulations for the bank to have a model that is not properly constrained (i.e. higher credit score corresponding to lower chance of approval). This is just one of many examples that show that it is significant to be able to impose not just monotonicity but any constraints known/required prior to training. Such techniques can also be useful when one can reasonably assume (rather than know for certain) that a feature should be constrained in a certain way.

---

### Official Review · AnonReviewer5 · 2020-11-06
**The paper provides a computationally efficient method for the modeling of monotonic functions. However, the authors should investigate various shape constraints, such as convexity, positivity, etc.**

**Rating:** 6
**Confidence:** 3

**Review:**

In this paper, the authors study statistical models based on the Kronecker-factored lattice (KFL) and investigated the condition that the monotonicity holds. The KFL provides efficient and flexible modeling for monotonic shape constraints. Furthermore, the ensemble of KFL is proposed, and its approximation ability is theoretically investigated. Some numerical experiments indicate that the KFL trains faster with fewer parameters with comparable prediction accuracy to existing methods.

This paper was assigned as an emergency review. So I did not have sufficient time to understand the content of this paper deeply.
The paper provides a computationally efficient method for the modeling of monotonic functions. However, the authors should investigate more variety of shape constraints such as convexity, positivity, etc.

Proposition 2 is an interesting result. However, more detailed studies on the shape would be necessary for the publication. For example, is it possible to derive the condition that the function is increasing in a variable and convex in the other variable?

Proposition 3 seems relatively straightforward. Supplementing untrivial statements would be good for readers to understand the significance of the proposition.

In the numerical experiments, the authors found that the hyperparameter V is important to tune the capacity of the proposed statistical model. Is there some theoretical support for that finding? Showing a clear insight would be excellent.

---

> ### Author Response · Authors · 2020-11-17
> **Response**
>
> We would like to thank the reviewer for their suggestions to improve our paper.
>
> 1. We have added proofs for both convexity and (what we believe is) positivity to the paper (see Section 3.2 and Appendix A). We also note that one can impose any number of these constraints on any dimensions so long as the conditions for each hold. However, we would like to clarify what the reviewer meant by the positivity shape constraint. If this is simply a nonnegative (or strictly positive) function, then it is straightforward to impose on KFL by enforcing nonnegativity (or positivity) on all parameters. We are happy to further revise the paper to update our definition if not correct. We believe that any more shape constraints beyond these are out of the scope of this paper, but we would love to explore even more shape constraints in future work, especially higher dimensional ones.
>
> 2. Can the reviewer clarify why they claim Proposition 3 is trivial? We note that the same proposition is not true when dealing with multilinear lattices. Namely, since $\mathcal{L}(\mathcal{V})$ is closed under addition, any function that is expressible as a sum of conventional multilinear lattices (with the same dimension and inputs) is also expressible as a single multilinear lattice; however, Proposition 5 (previously 3) implies that in general there are functions expressible as a sum of KFL functions which are not expressible using a single such lattice. We rephrased the proposition to make this detail more explicit.
>
> 3. We have added a section and corresponding proof to the paper showing how increasing $\mathcal{V}$ affects the model capacity (see Section 3.4 and Appendix A). We have also added more discussion using this proof as reasoning.

---

> > ### Comment · AnonReviewer5 · 2020-11-23
> > **response to the revision**
> >
> > Thanks for the thoughtful response.
> >
> > 1. I found that the authors addressed my concern in the revision.
> > 2. The wording of my comment was not sufficient. I did not think the past proposition 3 was trivial, but that the result sounded quite natural. I also understand a mathematically rigorous proof of such a result sometimes requires a somewhat involved argument. My concern was that the significance of their result was not very clear in the previous version. The revised expression of Proposition 5 is good for me.
> > 3. Thanks for the revision.
> >
> > The response to 1 is satisfactory for me, so I raise the score.

---

### Author Response · Authors · 2020-11-17
**Paper has been revised.**

Thank you to all four reviewers for taking the time to review our work. We really appreciate it! We responded to each reviewer individually and uploaded a revised copy of the paper incorporating the feedback we received.

---

### Author Response · Authors · 2021-01-21
**Camera ready version uploaded.**

Thank you so much to the reviewers and everyone involved for taking the time to review our paper thoroughly and provide good feedback. We have uploaded our camera ready submission. If you see anything that needs to be fixed that we missed, please say so and we'll update it.

---

### Decision · Program_Chairs · 2021-01-07
**Final Decision**

**Decision:**

Accept (Poster)

**Comment:**

The focus of the submission is shape-constrained regression, particularly the goal is to learn monotonic, 'reasonably rich' functions. In order to tackle this task, the authors extend the monotonic regression framework (Gupta et al., 2016) which scales less benignly in the input dimension. They propose to use lattice functions with parameters having Kronecker product structure (, and their ensembles). The resulting function class can be (i) stored and evaluated in linear time (Proposition 1), (ii) characterized / checked from monotonicity perspective (Proposition 2). The efficiency of the approach is demonstrated in three real-world examples.

Shape-constrained regression is a central topic in machine learning and statistics. The authors propose a parametric family to learn monotonically constrained functions. The storage and the evaluation of the resulting functions are both fast (linear), and the numerical experiments are encouraging. The submission can be of definite interest to the ICLR community.